# Ambient PM$_{2.5}$ exposure and expected premature mortality to 2100 in India under climate change scenarios

Sourangsu Chowdhury [ID] [1], Sagnik Dey[1] & Kirk R. Smith[2,3]

Premature mortality from current ambient fine particulate (PM$_{2.5}$) exposure in India is large, but the trend under climate change is unclear. Here we estimate ambient PM$_{2.5}$ exposure up to 2100 by applying the relative changes in PM$_{2.5}$ from baseline period (2001–2005) derived from Coupled Model Inter-comparison Project 5 (CMIP5) models to the satellite-derived baseline PM$_{2.5}$. We then project the mortality burden using socioeconomic and demographic projections in the Shared Socioeconomic Pathway (SSP) scenarios. Ambient PM$_{2.5}$ exposure is expected to peak in 2030 under the RCP4.5 and in 2040 under the RCP8.5 scenario. Premature mortality burden is expected to be 2.4–4 and 28.5–38.8% higher under RCP8.5 scenario relative to the RCP4.5 scenario in 2031–2040 and 2091–2100, respectively. Improved health conditions due to economic growth are expected to compensate for the impact of changes in population and age distribution, leading to a reduction in per capita health burden from PM$_{2.5}$ for all scenarios except the combination of RCP8.5 exposure and SSP3.

---

[1] Centre for Atmospheric Sciences, Indian Institute of Technology, Delhi, 110016, India. [2] School of Public Health, University of California Berkeley, Berkeley, CA 94720, USA. [3] Collaborative Clean Air Policy Centre Delhi, Delhi, 110003, India. Correspondence and requests for materials should be addressed to S.C. (email: sourangsu@cas.iitd.ac.in)

Long-term exposure to ambient particulate matter with aerodynamic diameter <2.5 μm (referred to as $PM_{2.5}$) has potential health risks due to cardiovascular and respiratory diseases leading to premature mortality[1–5]. India, the second-most populous country in the world, is recognized as a hotspot for aerosol loading[6] where ambient $PM_{2.5}$ exposure was observed to increase rapidly over the past decade (2001–2010)[7]. The annual premature mortality burden from ambient $PM_{2.5}$ exposure in India is currently estimated to be large by the Global Burden of Disease project[8,9] (~1.0 million), though such estimates are being updated due to modifications in concentration–response function based on new epidemiological evidence, alterations in concentration estimations due to different methodology and databases and adjustment for local health conditions. Thus the estimates should be taken as indicative only and are subject to modification over time[10]. Global warming has added further complexity in resolving this problem. Climate change is expected to modulate ambient $PM_{2.5}$ exposure by perturbing ventilation rate (that depends on atmospheric boundary layer depth and wind speed), precipitation scavenging, dry deposition, anthropogenic and natural emissions and background concentration[11]. Therefore, it is important to understand how the current ambient $PM_{2.5}$ concentration is projected to change in future under the warming climate[12] and how much the future exposure will translate into premature mortality burden depending on future population distribution and baseline mortality.

Whether $PM_{2.5}$ is projected to increase or decrease in the future as climate change proceeds is not clear[13,14]. Studies attempting to project the premature mortality burden due to ambient $PM_{2.5}$ exposure in the future are limited geographically and temporally. However, two of them examined conditions over America[15,16] and was restricted to 2050, one over Poland[17], one over East Asia[18] limited to 2020 and one global study extending up to the end of the century[19]. There is as yet no comprehensive study that quantifies the premature mortality burden attributed to projected change in $PM_{2.5}$ that incorporates demographic and socioeconomic data for a combination of pathways. A critical review by Madaniyazi et al.[20] recognized the urgency to project premature mortality burden due to ambient $PM_{2.5}$ exposure in developing countries for multiple scenarios rather than relying on a single model to realize the range of uncertainties in such estimate.

In the present study, we project the ambient $PM_{2.5}$ concentration in India by analysing 13 Coupled Model Inter-comparison Project 5 (CMIP5) model data for two climate change scenarios represented by representative concentration pathways, RCP4.5 and RCP8.5, and combine with demographic and socioeconomic projections used in the five shared socioeconomic pathway (SSP) scenarios as key scenario drivers[21,22] to estimate future premature mortality burden for 10 combined RCP–SSP scenarios. Ambient $PM_{2.5}$ exposure in India is projected to drop below the baseline period exposure only after 2050 under the RCP4.5 and after 2090 under the RCP8.5 scenario. Changes in meteorology induced by climate change are expected to decrease $PM_{2.5}$ exposure by 7–17% in the future under the RCP4.5 scenario. The baseline mortality is projected to decrease rapidly in future. On the contrary, population is projected to increase till 2050 for all five SSP scenarios beyond which it is projected to increase further only for SSP3. All these factors are expected to influence the projected premature mortality burden over India spatially and temporally. We hope that our results might help steer policy to mitigate air pollution in future decades such that co-benefits can be obtained by restricting both health impacts and climate change.

## Results

**Projection of ambient $PM_{2.5}$ exposure.** We project $PM_{2.5}$ exposure under RCP4.5 and RCP8.5 scenarios. We feel that the RCP2.6 scenario is difficult to achieve and the RCP6 scenario lies between RCP4.5 and RCP8.5. Therefore, RCP4.5 and RCP8.5 are expected to provide a realistic range of the estimates in future. RCP4.5 is a stabilization scenario where total radiative forcing is stabilized before 2100 at 4.5 W m$^{-2}$ without ever exceeding that value (650 ppm $CO_2$ equivalent) by employing a range of technologies and strategies for reducing greenhouse gas emissions[23]. RCP8.5 scenario corresponds to the pathway with the highest greenhouse gas emissions leading to a radiative forcing of 8.5 W m$^{-2}$ (1370 ppm $CO_2$ equivalent) by the end of the century[24].

We estimate the change in $PM_{2.5}$ exposure from 13 CMIP5 models (listed in Supplementary Table 1) relative to the baseline period (2001–2005) exposure. Simulations were carried out in 'historical' mode by CMIP5 models and also satellite-based $PM_{2.5}$ data are available for validation in the baseline period. Henceforth, we apply the mean relative change (across model ensemble) in $PM_{2.5}$ exposure to the satellite-derived $PM_{2.5}$ for the baseline period to project the $PM_{2.5}$ exposure for each pentad in future under the RCP4.5 and RCP8.5 scenarios. Ambient $PM_{2.5}$ exposure (estimated using approach 1 discussed in Methods section) averaged over the Indian landmass is estimated to be 34.5 μg m$^{-3}$ for the baseline period. CMIP5 model derived $PM_{2.5}$ shows good agreement with satellite-derived $PM_{2.5}$[7] for both the RCP scenarios for the period 2011–2015 (Supplementary Fig. 1) with a steeper slope and better (smaller intercept) agreement with the RCP8.5 scenario. In terms of bias (Supplementary Fig. 1), the ensemble of 13 CMIP5 models overestimates $PM_{2.5}$ compared to satellite-derived bias-corrected $PM_{2.5}$ over the dust source region of Rajasthan (marked by black box in Supplementary Fig. 1) and some parts of north India and underestimates over the rest of India as per projection under RCP4.5 scenario. Under the RCP8.5 scenario, the model ensemble bias is low (6–10%) in most parts of India except Rajasthan.

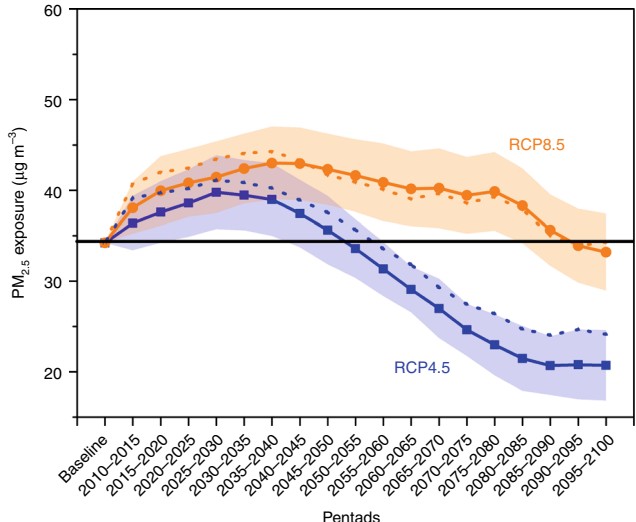

**Fig. 1** Projected ambient $PM_{2.5}$ exposure over the Indian landmass. Projected $PM_{2.5}$ exposure is averaged over India for future under both RCP4.5 and RCP8.5 scenarios. Shaded part represents the range (1–99%) in all-India averaged $PM_{2.5}$ across the 13 CMIP5 models. The black line represents the all-India averaged baseline (2001–2005) $PM_{2.5}$ exposure level. Solid and dashed lines indicate the projected exposure using approach 1 and approach 2, respectively.

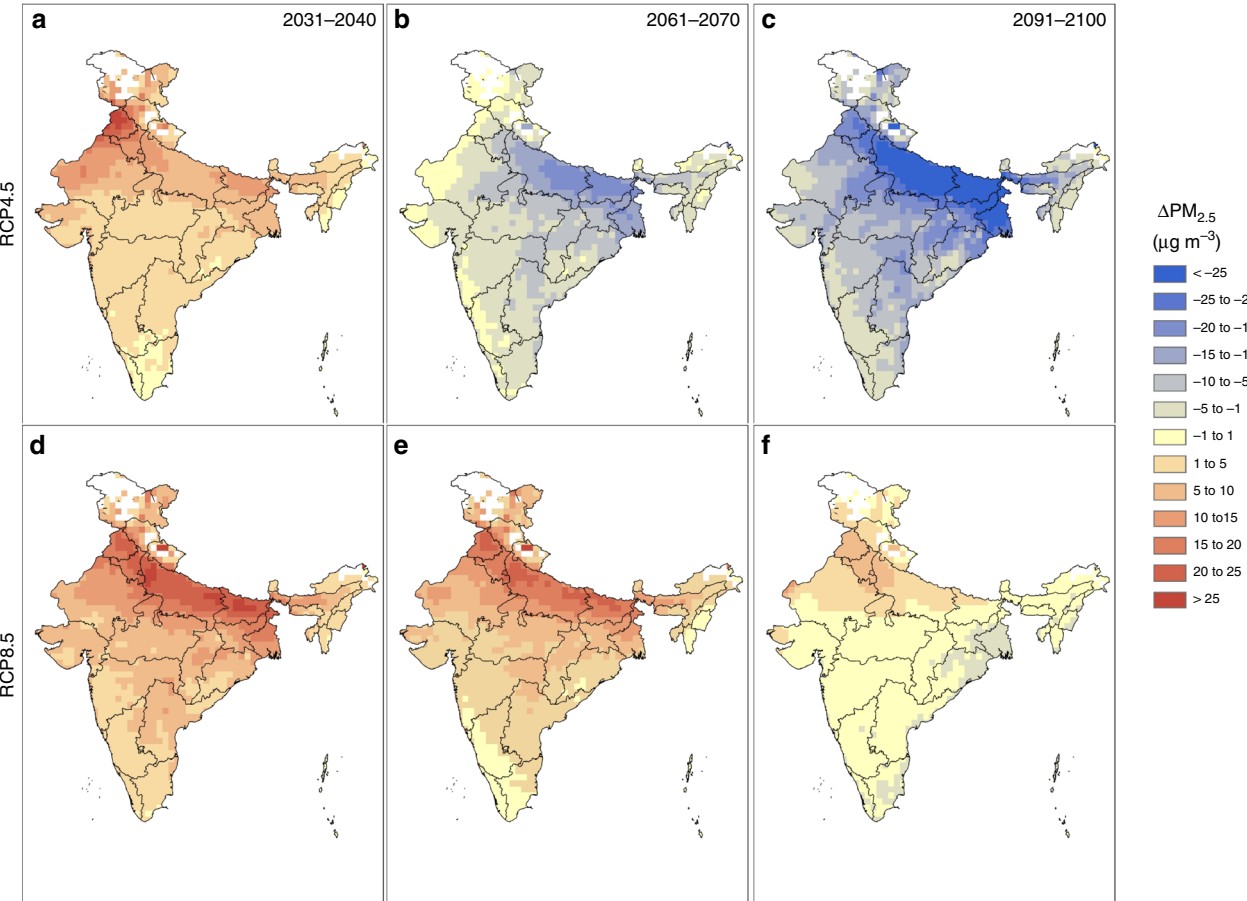

**Fig. 2** Projected spatial distribution of change in ambient $PM_{2.5}$ exposure with respect to baseline period over the Indian landmass. Spatial distribution of the projected changes in $PM_{2.5}$ exposure ($\Delta PM_{2.5}$) from the baseline (2001–2005) exposure under RCP4.5 for **a** near future (2031–2040), **b** distant future (2061–2070) and **c** far future (2091–2100) and under RCP8.5 scenario 5 for **d** near future (2031–2040), **e** distant future (2061–2070) and **f** far future (2091–2100) Reddish (bluish) tinge signifies projected increase (decrease) in $PM_{2.5}$ in future.

National ambient $PM_{2.5}$ exposure is projected to increase till 2030 followed by a decrease till the end of the century under the RCP4.5 scenario, while it is expected to increase till 2040 followed by a declining trend under the RCP8.5 scenario (Fig. 1). $PM_{2.5}$ exposure lowers below the 'baseline period' exposure only after 2090 under the RCP8.5 scenario. Under the RCP4.5 scenario, $PM_{2.5}$ exposure falls below the baseline period exposure after 2050 (Fig. 1). The shaded region in Fig. 1 depicts the range of $PM_{2.5}$ (mean $\pm 1\sigma$) obtained from 13 CMIP5 models across India. Changes in both meteorology and emission in future are expected to modulate the projected $PM_{2.5}$ exposure. The increase in $PM_{2.5}$ exposure in the initial decades under the RCP4.5 and RCP8.5 scenarios can be attributed to strong increase in emissions of $PM_{2.5}$ constituents that are considered in CMIP5 simulations, which are not directly due to climate change itself and could be modified by policy[25–27]. To isolate the impact of changing meteorology in warming climate from the impact of emission in modulating future $PM_{2.5}$ exposure, we apply the same $PM_{2.5}$/aerosol optical depth (AOD) ratio as of the baseline period (see approach 2 in the Methods section for more details). The difference between the projected $PM_{2.5}$ exposure using approach 2 and approach 1 estimates the contribution of climate change-induced meteorology in modulating $PM_{2.5}$ exposure.

Precipitation and wind speed are projected to increase over India under changing climate[28,29], which should result in a larger washout and dispersion of aerosols, thereby producing a decrease in surface $PM_{2.5}$ exposure. Increasing temperature and wind speed in a warming climate are expected to increase the mixing layer depth over India, which would also facilitate dispersion of $PM_{2.5}$ and thus reducing exposure. Under the RCP4.5 scenario, the effect of meteorology suppresses $PM_{2.5}$ exposure in the range ~7% in 2011–2015 to ~17% in 2095–2100 relative to the projections without considering the climate change-induced meteorological impact. Increase in emission strength[30] is expected to counteract the effect of meteorology in modulating $PM_{2.5}$ exposure after 2045 under RCP8.5 scenario except the last decade (2091–2100) when the meteorological impact is expected to overwhelm the effect of emissions once again. We average the $PM_{2.5}$ exposure of two succeeding pentads to estimate premature mortality burden at decadal scale.

Spatial heterogeneity of projected change of $PM_{2.5}$ concentrations relative to the baseline $PM_{2.5}$ exposure is depicted in Fig. 2, while the projected absolute values are shown in Supplementary Fig. 2. For the RCP4.5 scenario, $PM_{2.5}$ increases all over India in the decade 2031–2040, with the largest increase expected over the Indo-Gangetic Basin (Fig. 2a, warmer tinge signifies an increase in $PM_{2.5}$ in future, while bluish tinge signifies a decrease and yellow depicting regions with insignificant change). However, in the distant future (represented by the decade 2061–2070, Fig. 2b), $PM_{2.5}$ is projected to decrease all over India by 5–20 µg m$^{-3}$ and by a larger margin in the far future (represented by the decade 2091–2100, Fig. 2c). For the RCP8.5 scenario, $PM_{2.5}$ is expected to increase in parts of India (e.g., Indo-Gangetic Basin) up to the end of the century (Fig. 2d–f) by a greater margin than in the RCP4.5 scenario.

**Table 1 Projected mean estimates ($\pm$ uncertainty) of premature mortality burden per year (in million) in India due to ambient PM$_{2.5}$ exposure till the end of the century for the 10 combined RCP–SSP scenarios**

| Decade | SSP1 | | SSP2 | | SSP3 | | SSP4 | | SSP5 | |
|---|---|---|---|---|---|---|---|---|---|---|
| | RCP4.5 | RCP8.5 | RCP4.5 | RCP8.5 | RCP4.5 | RCP8.5 | RCP4.5 | RCP8.5 | RCP4.5 | RCP8.5 |
| 2011–2020 | 0.77 | 0.79 | 0.80 | 0.82 | 0.81 | 0.83 | 0.80 | 0.83 | 0.77 | 0.79 |
| | ($\pm$0.25) | ($\pm$0.25)[a] | ($\pm$0.26) | ($\pm$0.26) | ($\pm$0.27) | ($\pm$0.27) | ($\pm$0.26) | ($\pm$0.27) | ($\pm$0.25) | ($\pm$0.26) |
| 2021–2030 | 0.67 | 0.69 | 0.73 | 0.75 | 0.80 | 0.82 | 0.76 | 0.78 | 0.65 | 0.67 |
| | ($\pm$0.21) | ($\pm$0.22)[a] | ($\pm$0.23) | ($\pm$0.24) | ($\pm$0.26) | ($\pm$0.26) | ($\pm$0.24) | ($\pm$0.25) | ($\pm$0.21) | ($\pm$0.21) |
| 2031–2040 | 0.62 | 0.64 | 0.70 | 0.72 | 0.83 | 0.85 | 0.75 | 0.78 | 0.57 | 0.59 |
| | ($\pm$0.19) | ($\pm$0.20)[a] | ($\pm$0.22) | ($\pm$0.23) | ($\pm$0.26) | ($\pm$0.27) | ($\pm$0.24) | ($\pm$0.25) | ($\pm$0.18) | ($\pm$0.18) |
| 2041–2050 | 0.56 | 0.59 | 0.66 | 0.70 | 0.84 | 0.89 | 0.73 | 0.77 | 0.50 | 0.53 |
| | ($\pm$0.17) | ($\pm$0.18)[a] | ($\pm$0.21) | ($\pm$0.22) | ($\pm$0.27) | ($\pm$0.28) | ($\pm$0.23) | ($\pm$0.24) | ($\pm$0.15) | ($\pm$0.16) |
| 2051–2060 | 0.47 | 0.52 | 0.59 | 0.65 | 0.84 | 0.93 | 0.66 | 0.73 | 0.41 | 0.45 |
| | ($\pm$0.15) | ($\pm$0.16)[a] | ($\pm$0.19) | ($\pm$0.20) | ($\pm$0.27) | ($\pm$0.30) | ($\pm$0.21) | ($\pm$0.23) | ($\pm$0.13) | ($\pm$0.14) |
| 2061–2070 | 0.39 | 0.46 | 0.50 | 0.58 | 0.80 | 0.94 | 0.55 | 0.66 | 0.32 | 0.37 |
| | ($\pm$0.13) | ($\pm$0.14)[a] | ($\pm$0.16) | ($\pm$0.18) | ($\pm$0.27) | ($\pm$0.30) | ($\pm$0.18) | ($\pm$0.21) | ($\pm$0.10) | ($\pm$0.11) |
| 2071–2080 | 0.30 | 0.39 | 0.40 | 0.52 | 0.73 | 0.94 | 0.44 | 0.56 | 0.24 | 0.30 |
| | ($\pm$0.10) | ($\pm$0.12)[a] | ($\pm$0.14) | ($\pm$0.16) | ($\pm$0.26) | ($\pm$0.30) | ($\pm$0.15) | ($\pm$0.18) | ($\pm$0.08) | ($\pm$0.09) |
| 2081–2090 | 0.23 | 0.31 | 0.33 | 0.45 | 0.68 | 0.92 | 0.34 | 0.46 | 0.17 | 0.24 |
| | ($\pm$0.09) | ($\pm$0.97)[a] | ($\pm$0.13) | ($\pm$1.14) | ($\pm$0.26) | ($\pm$0.30) | ($\pm$0.13) | ($\pm$0.15) | ($\pm$0.06) | ($\pm$0.07) |
| 2091–2100 | 0.18 | 0.25 | 0.28 | 0.37 | 0.66 | 0.88 | 0.28 | 0.37 | 0.14 | 0.18 |
| | ($\pm$0.07) | ($\pm$0.08)[a] | ($\pm$0.11) | ($\pm$0.12) | ($\pm$0.27) | ($\pm$0.29) | ($\pm$0.11) | ($\pm$0.12) | ($\pm$0.05) | ($\pm$0.06) |

[a] Combination of RCP8.5 scenario and SSP1 population is practically impossible because the storyline of RCP8.5 involves high greenhouse gas emission and SSP1 is narrated to be a green-growth paradigm

**Projection of premature mortality burden**. Although the RCP scenarios incorporate a combination of socioeconomic parameters such as population and income[31,32], it can be argued that a given emission pathway defined by each of the RCP scenarios may be reached under a wide variety of socioeconomic conditions[33]. The RCP8.5–SSP1 combination is not feasible[34]; therefore, projected premature mortality burden is discussed only for the remaining nine RCP–SSP combinations.

Supplementary Fig. 3 describes the SSPs spanning the space of challenges to mitigation and adaptation to climate change[35,36] (see Supplementary Note). As we move away from the origin diagonally (from SSP1 towards SSP3), challenges for both mitigation of climate change and adaptation to it increase. SSP2 is the midway scenario with intermediate challenges for adaptation and mitigation, and SSP4 is the scenario with no challenge for mitigation but a higher challenge for adaptation. SSP5 has high challenge for mitigation but no challenge for adaptation. We note that the SSP forcing targets are not considered in this analysis; rather we use the RCP scenarios to estimate PM$_{2.5}$ exposure. Further, projections of population and gross domestic product (GDP)[22] that were used as demographic and socioeconomic drivers to define the five SSP scenarios are used in combination with the exposure based on two RCP scenarios to estimate the premature mortality burden (see Methods section for mathematical depiction).

We estimate the future relative risk (RR) from ambient PM$_{2.5}$ exposure utilizing the integrated exposure–response (IER) function[37]. We adjust baseline mortality spatially and temporally using functions in which baseline mortality decreases non-linearly with projected GDP[10] that varies across the five SSP scenarios[22] (Supplementary Fig. 4). The shaded portion in Supplementary Fig. 4 shows the uncertainty in terms of the model coefficients as described in our previous work[10]. We then develop 10 climate change–socioeconomic scenarios by combining the two RCP scenarios and socioeconomic and demographic drivers used in the five SSP scenarios. Population projection data are taken from demographic drivers used by the quantitative models in developing SSP scenarios[34] (see Supplementary Note). We provide detailed statistics of premature death from ambient

PM$_{2.5}$ exposure at country level for all the decades (Table 1) and at state level (see Supplementary Dataset) for the three representative decades 2031–2040 (near future), 2061–2070 (distant future) and 2091–2100 (far future) with the discussion focussing on these three representative decades.

Figure 3 shows the future premature mortality burden in India relative to the baseline period. Under the RCP4.5 scenario for all five demographic and socioeconomic projections of the SSP scenario drivers, total premature mortality burden is expected to decrease in the future relative to the baseline period (Fig. 3a and Table 1). We speculate that the reason for this is the steep projected fall in the baseline mortality for all the diseases in the future (Supplementary Fig. 4) as the GDP is expected to increase over India as per the SSP scenario descriptions. Crude premature mortality rate (per 100,000 exposed population) is also expected to decrease in future for all the SSP population projections under RCP4.5 scenario (Supplementary Table 2). Highest ($\pm 1\sigma$) rate of crude premature mortality burden per 100,000 exposed population at the end of the century is expected for the SSP3 population ($11.8 \pm 4.8$) followed by SSP4 ($7.3 \pm 2.9$), SSP2 ($5.8 \pm 2.4$), SSP1 ($4.3 \pm 1.7$) and SSP5 ($3.2 \pm 1.3$) for the RCP4.5 scenario. The combination of SSP3 and RCP8.5 scenario projects premature mortality burden to increase above the baseline period after 2050. This is because SSP3 population is projected to increase rapidly in the future decades, which in combination with a higher ambient PM$_{2.5}$ exposure under RCP8.5 is expected to overcome the reduction in baseline mortality per capita, thus aggravating the premature mortality burden. Combinations of all other SSP demography and socioeconomic projections with PM$_{2.5}$ exposure for RCP8.5 scenario project a decrease in premature mortality burden in future (Fig. 3b and Table 1). Premature mortality burden per 100,000 population (Supplementary Table 2) for the combination of SSP3 population under the RCP8.5 scenario exposure decreases throughout the century relative to the baseline period. This further clarifies the claim that the projected surge in SSP3 projected population enhances the premature mortality burden in the middle of the century above baseline period premature mortality burden (Fig. 3). Crude premature mortality burden per 100,000 population at the end of the century under

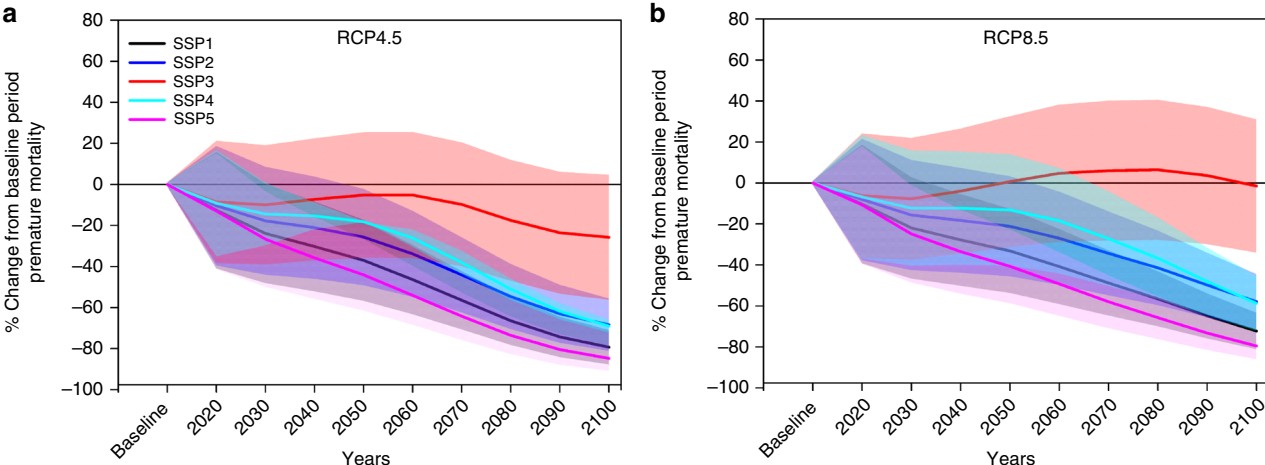

**Fig. 3** Projected change in premature mortality due to PM$_{2.5}$ exposure w.r.t. baseline period over the Indian landmass. Projected percentage changes in premature mortality burden from ambient PM$_{2.5}$ exposure in India for the five SSP scenario population using projected baseline mortality under **a** RCP4.5 and **b** RCP8.5 scenarios. The range (1–99%) of premature mortality (as a function of standard error in baseline mortality model and the range of PM$_{2.5}$ from CMIP5 models) is shown as shades around the mean values as bold lines.

**Table 2 Framework of the sensitivity study where the upward and downward arrows indicate an increase and decrease of the respective parameter relative to the baseline period in that particular sensitivity study while 'no change' is indicated by horizontal arrow**

| | PM$_{2.5}$ | Baseline Mortality | Population | Interpretation |
|---|---|---|---|---|
| SA1 | ↔ | ↔ | ↑ | This scenario isolates the change in premature mortality burden in future only due to demographic transition. It is estimated by using population distribution of the respective decade and baseline period PM$_{2.5}$ and baseline mortality. The difference between this estimate and baseline period premature mortality burden estimate indicates the sole contribution of demographic transition |
| SA2 | ↔ | ↑ | ↔ | This scenario isolates the change in premature mortality burden in future only due to epidemiological transition. It is estimated by using baseline mortality of the respective decade and baseline period PM$_{2.5}$ and population distribution. The difference between this estimate and baseline period premature mortality burden estimate indicates the sole contribution of demographic transition |
| SA3 | ↑[a] | ↑ | ↑ | This scenario quantifies the change in premature mortality burden in future due to the changes in meteorology under climate change scenarios. It is quantified by the difference between premature mortality burden estimate from PM$_{2.5}$ exposure by approach 2 and between premature mortality projected by using PM$_{2.5}$ exposure estimates from approach 1. (See Methods section for more details) |
| SA4 | ↓[b] | ↑ | ↑ | This scenario quantifies the possible averted premature mortality if policy intervention to reduce PM$_{2.5}$ exposure is enforced such that average PM$_{2.5}$ exposure over the Indian landmass meets WHO-IT1 (35 µg m$^{-3}$) by near future, WHO-IT2 (25 µg m$^{-3}$) in distant future and WHO-IT3 (15 µg m$^{-3}$) in far future |

The purpose of each sensitivity study and its interpretation are provided in the last column
↑[a]PM$_{2.5}$ is considered changing but is estimated by using approach 2 (see Methods section)
↓[b]PM$_{2.5}$ is considered changing by policy interventions restricting PM$_{2.5}$ in targeted grids

RCP8.5 scenario is expected to be highest for SSP3 population (15.6 ± 5.2) followed by SSP 4 (9.6 ± 3.1), SSP2 (7.7 ± 2.5) and SSP5 (4.3 ± 1.4). The total premature mortality burden for all the 10 combined scenarios in all the decades is summarized in Table 1.

The key observations are summarized as follows. Premature mortality is expected to decrease in future decades compared to the baseline period. The highest premature mortality burden is expected for the SSP3–RCP8.5 combined scenario in all the decades, whereas lowest premature mortality is expected for the SSP5–RCP4.5 combined scenario. Premature mortality burden is expected to be 2.4–4, 9.7–17.9 and 28.5–38.8% higher under RCP8.5 scenario relative to RCP4.5 scenario in near future, distant future and far future, respectively. Across all the five population distributions following SSP scenarios, annual mean

(± 1σ) crude premature mortality burden per 100,000 population from ambient PM$_{2.5}$ exposure in India is estimated as 18.1 ± 4.6 in near future (2031–2040), 10.5 ± 3.5 in distant future (2061–2070) and 6.5 ± 2.6 in far future (2091–2100), respectively, under the RCP4.5 scenario. The corresponding estimates under the RCP8.5 scenario are 18.7 ± 5.9, 12.3 ± 3.9 and 8.6 ± 2.8, respectively, in the near, distant and far future.

The shaded areas in Fig. 3 represent the range of uncertainty in premature mortality estimates across the country. The upper and lower ranges around the mean values are estimated using the mean ± 1σ PM$_{2.5}$ exposure (where σ is the error) for each grid obtained from 13 CMIP5 models and the mean ± 1σ values of the coefficients used in the GDP–baseline mortality relation[10]. In contrast to a recent study[19] that projects a decrease in premature mortality burden below the present-day level after 2050 in India

**Table 3 Changes in projected premature mortality burden for the four sensitivity studies (SA) relative to the burden in the baseline period. Values are in 1000s. The details of the four sensitivity studies are summarized in Table 2**

| Decades | SSP1 | | SSP2 | | SSP3 | | SSP4 | | SSP5 | |
|---|---|---|---|---|---|---|---|---|---|---|
| | RCP4.5 | RCP8.5 | RCP4.5 | RCP8.5 | RCP4.5 | RCP8.5 | RCP4.5 | RCP8.5 | RCP4.5 | RCP8.5 |
| **SA1** | | | | | | | | | | |
| 2031–2040 | 792.8 | | 784.6 | | 786.2 | | 749.3 | | 786.1 | |
| 2061–2070 | 1021.8 | | 1127.9 | | 1241.6 | | 955.7 | | 1002.1 | |
| 2091–2100 | 612.5 | | 980.2 | | 1435.3 | | 548.4 | | 588.9 | |
| **SA2** | | | | | | | | | | |
| 2031–2040 | −575.8 | | −532.8 | | −471.2 | | −498.7 | | −599.3 | |
| 2061–2070 | −687.1 | | −644.7 | | −516.5 | | −592.5 | | −721.6 | |
| 2091–2100 | −737.4 | | −706.1 | | −547.2 | | −655.5 | | −775.9 | |
| **SA3** | | | | | | | | | | |
| 2031–2040 | −9.1 | −5.9[a] | −10.3 | −6.7 | −12.3 | −7.9 | −11.1 | −7.2 | −9.1 | −5.5 |
| 2061–2070 | −19.2 | 4.5[a] | −24.5 | 5.7 | −39.5 | 9.3 | −27.2 | 6.5 | −15.7 | 3.6 |
| 2091–2100 | −25.6 | −0.8[a] | −38.1 | −1.3 | −88.4 | −2.9 | −37.5 | −1.2 | −18.8 | −0.7 |
| **SA4** | | | | | | | | | | |
| 2031–2040 | −28.1 | −44.8 | −31.9 | −50.9 | −37.7 | −60.3 | −34.4 | −54.8 | −25.8 | −41.1 |
| 2061–2070 | −25.2 | −83.1 | −32.1 | −105.5 | −51.9 | −170.3 | −35.7 | −117.4 | −20.7 | −64.9 |
| 2091–2100 | −41.7 | −89.5 | −62.6 | −132.9 | −145.7 | −308.4 | −61.1 | −129.9 | −30.8 | −66.1 |

[a] The combination of RCP8.5 and SSP1 scenario is practically impossible because the storyline of RCP8.5 involves high greenhouse gas emission and SSP1 is narrated to be a green-growth paradigm

only for the RCP4.5 scenario, our results depict a decrease in the premature mortality burden below the present-day level much earlier by the next decade under both the RCP4.5 and RCP8.5 scenarios except the RCP8.5–SSP3 combined scenario. This difference can be attributed to our consideration of the projected improving economic condition in India as a major determining factor in our estimated premature mortality burden in future.

**Attribution of premature mortality burden to key factors.** Premature mortality burden from ambient $PM_{2.5}$ exposure depends on baseline mortality and exposed population. We carry out four sensitivity studies (SA1–SA4; summarized in Table 2) to understand the relative importance of each of these individual factors in the estimated burden. In the first sensitivity study (SA1), we isolate the impact of future demographic transition on the premature mortality burden. The second sensitivity study (SA2) isolates the impact of epidemiologic/health transition, due to the growing economy, on the future burden. The third sensitivity study (SA3) is designed to separate out the impact of meteorological changes due to climate change on the future premature mortality burden. In the fourth sensitivity study (SA4), we assume that policy interventions would be enforced on sources of air pollution, which would limit the $PM_{2.5}$ exposure in India to meet recommended air quality guidelines.

We estimate the expected premature death in 3 representative decades (2031–2040, 2061–2070 and 2091–2100) for two hypothetical cases. In the first case, we use projected population based on SSP scenarios but the baseline mortality rate and $PM_{2.5}$ exposure are considered for the baseline period. The difference between this case and the estimated premature mortality using baseline period exposure, exposed population and baseline mortality can be attributed solely to the demographic transition and is tagged as SA1 (Table 3). The positive values indicate that projected demographical change in the future decades plays a role in aggravating the premature mortality burden in future. We see that the contribution of demography towards modulating the premature mortality burden is the highest in the distant future (2061–2070) for SSP1, SSP2, SSP4 and SSP5 when the projected population for these four SSP scenarios reach the peak (Supplementary Fig. 5). On the contrary, the demographic

contribution is expected to be the highest in the last decade (2091–2100) for SSP3 scenario. The projected change in Indian population by age is depicted in Supplementary Fig. 6. For all the SSP scenarios, we see that the aged population is projected to increase substantially while the relative count of younger population is projected to decrease.

In the second case, we use population and exposure for the baseline period and the baseline mortality as a function of projected GDP in future from all the SSP scenarios to estimate the premature mortality burden. The difference between this case and the estimated premature mortality using baseline period exposure, exposed population and baseline mortality can be attributed solely to the epidemiologic transition and is labelled as SA2 (Table 3). The negative change indicates that epidemiologic transition in future is expected to reduce the premature mortality burden. It is noted that the demographic and epidemiologic changes act in opposite direction though negative epidemiologic transition is unable to compensate fully for positive demographic transition for any of the SSP populations in any of the representative decades except for SSP1 and SSP5 in far future.

We also perform an analysis to examine the effect of changing meteorology due to climate change in modulating the projected premature mortality burden (SA3). We project the difference in premature mortality estimated by using $PM_{2.5}$ exposure following approach 2 and $PM_{2.5}$ exposure following approach 1 for both the RCP scenarios (Table 3). The positive and negative values indicate that climate change-induced meteorology is expected to avert and aggravate premature mortality burden, respectively, in future. A previous study[38] estimated a little benefit in terms of averted premature mortality in South Asia till 2030 due to projected meteorological changes due to climate change. However, for India, our estimate is that meteorology will help averting 9100–12,300 premature deaths per year in near future (2031–2040) under the RCP4.5 scenario and 5500–7900 premature deaths under RCP8.5 scenario for all the SSP populations. In distant future, favourable meteorological condition is projected under the RCP4.5 scenario to avert 15,700–39,500 premature deaths per year. In the far future, 18,800–88,400 premature deaths per year will be averted under the RCP4.5 scenario due to meteorological changes induced by global warming. It is expected that the meteorology under RCP8.5 scenario will aggravate premature mortality burden per year after 2050 until the

last decade of the century when the meteorological condition turns favourable to avert 700–2900 deaths per year.

The projected premature mortality burden in India may change if air pollution mitigation measures are implemented in future. We perform a sensitivity (referred to as SA4, Table 3) analysis to quantify averted premature mortality in such case. There are various possible pathways for policy implementation in the future that are difficult to foresee. We assume that air pollution mitigation policies will be enforced over the entire Indian landmass that would result in a reduction of $PM_{2.5}$ exposure such that the all India average $PM_{2.5}$ exposure in the respective decades meets World Health Organization (WHO) interim target (IT)-1, 2 and 3 standards in near future, far future and distant future, respectively. If average ambient $PM_{2.5}$ exposure in India meets WHO IT-1 (i.e., 35 µg m$^{-3}$) by 2031–2040, about 28,000–38,000 premature deaths can be averted under the RCP4.5 scenario and 41,100–60,100 under the RCP8.5 scenario every year across all the projected populations. Similarly if mitigation policies are further enforced such that average exposure to $PM_{2.5}$ in India meets WHO IT-2 (25 µg m$^{-3}$) by 2061–2070, 20,800–51,900 premature deaths can be averted under the RCP4.5 scenario and about 64,900–170,500 under the RCP8.5 scenario every year. We further assume that if all India average $PM_{2.5}$ exposure meets WHO IT-3 (15 µg m$^{-3}$) in the last decade of century with implementation of stringent measures, every year 30,800–145,800 lives can be saved under the RCP4.5 scenario and 66,000–308,500 under the RCP8.5 scenario. Achieving these targets may be difficult given the existing high level of ambient $PM_{2.5}$ exposure (about twice or more than the Indian annual standard) in the Indo-Gangetic Basin. Even if India does not meet these targets by the specified years, any progress towards achieving these targets will have substantial health benefits.

## Discussion

We project premature mortality burden from ambient $PM_{2.5}$ exposure in India for the period 2011–2100. Our results provide an overview of premature mortality burden attributed to demographic and epidemiologic changes and meteorological changes in response to climate change. Though the premature mortality burden from $PM_{2.5}$ exposure in India is projected to decrease in the forthcoming decades, the absolute values are quite large and therefore the decreasing trend should not be taken lightly. Our estimates (absolute numbers) vary due to the underlying assumptions (see Methods section), but banded estimates that incorporate variations in assumptions provide comprehensive datasets across multiple scenarios that will facilitate policymakers in formulating appropriate strategy to deal with climate change in India. We note that, in addition to change in ambient $PM_{2.5}$, projected temperature rise in the future decades may propel formation of ozone[39] and secondary particles, which may increase the premature mortality burden due to air pollution in future. This issue is beyond the scope of the present work and requires detailed analysis in future.

The coverage of the Great Pollution Episode of 2016 in Delhi by the mass media and the reports of the Global Burden of Disease (GBD) assessment have created an awareness among the general public about the menace of $PM_{2.5}$ on human health in India. After the November episode, graded action plan was implemented in the city of Delhi by the Ministry of Environment, Forest and Climate Change as per the directive of Hon'ble Supreme Court of India (http://envfor.nic.in/content/so-118e-12-01-2017-graded-response-action-plan-control-air-pollution-delhi-and-national-capi). The Ministry of Health and Family Welfare, Government of India has also started various

initiatives[40] including formation of a steering committee that published a report on a multi-sectoral approach to combat air pollution. One can hope that these efforts will drive intersectoral mitigation measures in the future to create a cleaner India just like the policy interventions after the Great London Smog event of 1952 succeeded in making London one of the cleanest cities in the world[41]. India, however, is coming to recognize the need for air pollution control when it still has much traditional pollution in villages due to biomass fuel use as well as a growing and largely uncontrolled modern sector. Both have significant impacts and interact in the sense that household sources are also major sources of ambient pollution. As a result, the total health burden per capita is likely higher in major Indian cities (e.g., Delhi) than in London, which started its control of ambient pollution well after it had given up open fires for cooking, although still using coal for space heating. Arguably, therefore the urgency is higher now in India. The Lancet Commission 2015[42] identifies that the continued acceleration of greenhouse gases and pollutant emissions along with changing demographical features would make climate change an increasingly severe risk to human health. Implementing measures to mitigate air pollutants like $PM_{2.5}$ assures co-benefits in terms of reducing climate altering pollutants (which plays an important role in accelerating climate change) and averting premature mortality[43].

One option for national policy could be to follow SA4 (see Table 2) so that India achieves WHO IT-1, 2 and 3 in the near, distant and far future, respectively. It is important to note that, if India and rest of the world follow the RCP8.5 scenario, more stringent policy measures are required (relative to RCP4.5) to meet the targets discussed in SA4 including strong control of household sources. Any other pathways to meet air quality targets may modulate the projected burden differently in future. Although the relationship between climate change and air pollution is not a simple one, the country needs to take strong actions to reduce both kinds of impacts if health risks are to be tempered. If climate changes at a faster pace, it is even more important to reduce emission if the public's health is to be protected.

## Methods

**Satellite-derived $PM_{2.5}$ for the baseline period.** Dearth of systematic $PM_{2.5}$ measurement over India prompted us to exploit use of the satellite aerosol products to derive $PM_{2.5}$ for this study following our earlier studies[7,10]. Multiangle Imaging SpectroRadiometer (MISR)-retrieved daily columnar AOD is converted to $PM_{2.5}$ using a spatio-temporally varying conversion factor ($\eta$), which may also be expressed as the ratio of surface $PM_{2.5}$ and AOD and is simulated by GEOS-Chem model[44] with aerosol vertical distribution constrained by CALIOP (Cloud Aerosol Lidar and Infrared Satellite Observations) measurements[45,46]. Satellite-derived $PM_{2.5}$ was bias corrected[10] using coincident in situ observations. 2001–2005 is considered as the 'baseline period' in our study in line with the CMIP5 model simulation strategy where the CMIP5 models are simulated in historical mode up to 2005, and after 2005, the models are simulated under RCP scenarios.

**Model-derived $PM_{2.5}$.** Thirteen models from the CMIP5 family were considered in this study, name of which along with their respective grid resolutions are listed in Supplementary Table 1. $PM_{2.5}$ was estimated from each of the models for the period 2011 up to 2100 for RCP4.5 and RCP8.5 scenarios following two approaches. Estimated $PM_{2.5}$ by each model was downscaled statistically to 0.5° × 0.5° resolution to negate the inconsistency among the grid resolutions of these 13 models and to be at par with the resolution of MISR-retrieved $PM_{2.5}$.

Approach 1: $PM_{2.5}$ is estimated as a sum of the mass of individual species from each of the 13 CMIP5 model:

$$PM_{2.5} = BC + OA + SO_4 + NH_4 + (0.25 \times SS) + (0.1 \times dust) \qquad (1)$$

where BC is black carbon, OA is organic aerosols and SS is sea salt. Total OA can be expressed as the sum of POA (primary organic aerosol) and SOA (secondary organic aerosol)[47,48]. While all the models report BC, $SO_4$, SS and dust, 8 models report OA, 2 models report only SOA and 2 models report only POA, 1 model (GFDL-CM3) reports both SOA and POA. To estimate SOA for those models that report only POA, the ratio of POA and SOA obtained from GFDL-CM3 is applied.

Same technique is applied for estimating POA for those models that only informs about SOA. Surface concentration of $NH_4$ is not reported by any of the 13 CMIP5 models and so it is estimated as $NH_4 = (36 \times SO_4)/96$ assuming that $NH_4$ is only present as ammonium sulphate. We follow a study[49] in estimating the fraction of dust and SS to fall within the $PM_{2.5}$ size fraction. We assume the fraction to remain uniform throughout the Indian landmass in the absence of information about spatial heterogeneity.

Approach 2: AOD simulated by each of the 13 CMIP5 models was converted to surface $PM_{2.5}$ using the same conversion factor $\eta$ (i.e., $PM_{2.5}$/AOD) as has been used to estimate satellite-derived $PM_{2.5}$ for the baseline period. Since the space–time variation of $\eta$ depends on meteorology, consideration of the same $\eta$ in the future where AOD is simulated from changing emissions as per RCP scenarios implies that the estimated $PM_{2.5}$ in future is only governed by the change in emission and not by the meteorology. Difference in the estimated $PM_{2.5}$ by these two approaches can isolate the role of meteorology in warming climate in modulating $PM_{2.5}$ distribution.

The CMIP5 models are known to underestimate (up to a factor of 10) AOD and the $PM_{2.5}$ components over the Indian subcontinent[50,51]. To account for this underestimation, we estimate the relative changes in $PM_{2.5}$ for each pentad (starting from 2011 to 2100) from the baseline period $PM_{2.5}$ estimated by the CMIP5 models. We apply back the relative changes to the baseline period satellite-derived exposure to estimate the future $PM_{2.5}$ exposure at $0.5° \times 0.5°$ grid. The procedure can be represented by the following equation (Eq. 2):

$$PM_{2.5^{mod}_{pentad=x}} = PM_{2.5^{sat}_{pentad=bl}} + PM_{2.5^{sat}_{pentad=bl}} \times \left[ \frac{PM_{2.5^{model}_{pentad=x}} - PM_{2.5^{model}_{pentad=bl}}}{PM_{2.5^{model}_{pentad=bl}}} \right] \quad (2)$$

Here $PM_{2.5^{mod}_{pentad=x}}$ represents $PM_{2.5}$ for a given pentad $x$ in the future that is eventually used for further analysis of premature mortality burden (as per Eqs. 3 and 4 in Methods section). $PM_{2.5^{sat}_{pentad=bl}}$ is the MISR-derived $PM_{2.5}$ for the baseline (bl) period (2001–2005). $PM_{2.5^{model}_{pentad=x}}$ is the CMIP5 model-derived $PM_{2.5}$ (as per Eq. 1 in the Methods section). $PM_{2.5^{model}_{pentad=bl}}$ is the CMIP5 model-derived $PM_{2.5}$ for the baseline period.

**RR and premature mortality burden.** A non-linear IER[37] function developed recently constrains the shape of the concentration response function. The IER estimates RR across a wide range of exposure including ambient air pollution, household air pollution, active smoking and second-hand smoking. We choose four diseases, chronic obstructive pulmonary disease (COPD), ischaemic heart disease (IHD), stroke and lung cancer (LC). These diseases have direct causal links to ambient $PM_{2.5}$ exposure[1,52,53]. The IER function can be expressed as in Eq. 3

$$RR_{i,j} = 1 + \alpha_j[1 - \exp(-\gamma_j(\Delta PM_{2.5})_i^{\delta_j})] \quad (3)$$

where $RR_{i,j}$ represents the RR for a disease $j$ at a grid $i$ for a specific decadal exposure to ambient $PM_{2.5}$. $(\Delta PM_{2.5})_i$ denotes the change in $PM_{2.5}$ exposure from the counterfactual $PM_{2.5}$ exposure of 5.8 µg m$^{-3}$ as used in GBD2010[54] and $\alpha_j, \gamma_j$ and $\delta_j$ are specific constants for each of the diseases. We use the median values of RR for each disease (for IHD and stroke, age-specific values of RR have been considered), which are provided as a look-up table by a previous study[55]. More details about the IER function can be found elsewhere[37].

Premature mortality per year ($\Delta M_{i,j}$) due to a disease $j$ for each $0.5°$ grid $i$ attributable to ambient $PM_{2.5}$ exposure has been estimated for each decade starting from 2011 to 2100 using the traditional epidemiological relation[10]. as in Eq. 4

$$\sum_{i,j=1}^{N} \Delta M_{i,j} = \sum_{i,k,j=1}^{N} y_{i,k,j} \times \frac{\sum_{i=1}^{N} RR_{i,j} - 1}{\sum_{i=1}^{N} RR_{i,j}} \times \sum_{i=1}^{N} P_i \quad (4)$$

In this equation, $y_{i,j,k}$ is the baseline mortality for a disease $j$ in a grid $i$ located geographically within a state $k$. Exposed population for each grid for the five SSP scenarios has been obtained from the Inter-Sectoral Impact Model Comparison Project (http://clima-dods.ictp.it/Users/fcolon_g/ISI-MIP). The population used as drivers in these five SSP scenarios are defined as 'reference' scenarios, because they assume no change in climate policies[32,33]. The population in each of the five SSP scenarios (SSP1–5)[56–60] defines different socioeconomic challenges to mitigation and adaptation strategies to deal with climate change (more details in Supplementary Note). Age-wise demographic categorization was available at national level for each of the scenarios. For the exposure assessment, the adult (age >25 years) population ($P_i$) was considered. $P_i$ for decadal exposure to $PM_{2.5}$ was considered to be the population in the first year of the respective decade. For example, population of 2021 is used for calculating premature mortality burden due to ambient $PM_{2.5}$ exposure for the decade 2021–2030, because it represents the minimum population that gets exposed to decadal $PM_{2.5}$.

Age-specific population for each grid was estimated as in Eq. 5:

$$pop^{grid=z,SSP=w}_{decade=x,age\,group=y} = f_{pop^{national,SSP=w}_{decade=x,age\,group=y}} \times pop^{grid=z,SSP=w}_{total\,decade=x} \quad (5)$$

where $pop^{grid=z,SSP=w}_{decade=x,age\,group=y}$ is the total population for an age slab '$y$' for a particular $0.5° \times 0.5°$ grid '$z$' in a decade '$x$' and for a SSP scenario '$w$'. $f_{pop^{national,SSP=w}_{decade=x,age\,group=y}}$ is the

fraction of population in a particular age group '$y$' for India obtained for a SSP scenario '$w$' from https://tntcat.iiasa.ac.at/SspDb/dsd?Action=htmlpage&page=about and $pop^{grid=z,SSP=w}_{total\,decade=x}$ represents total population for each grid. Age-specific population database is required to estimate age-specific RR for IHD and stroke. For COPD and LC, total population ($P_i$) aged >25 years is used following Eq. 6:

$$P_i = \sum_y pop^{grid=z,SSP=w}_{decade=x,age\,group=y} \quad (6)$$

We adjust the baseline mortality for each of the diseases as a function of GDP (see our previous study[10] for more information on the non-linear functions). GDP-PPP (purchasing power parity) is obtained at national level from https://tntcat.iiasa.ac.at/SspDb/dsd?Action=htmlpage&page=countries for two sets of simulations— IIASA GDP_v9_130219 and OECD Env-Growth_v9_130325—for all the five SSP scenarios. GDP-PPP from both these sets for each decade is averaged. Rate of change of GDP-PPP for all the decades up to 2100 relative to the baseline period GDP-PPP (which is considered to be 2010 for India obtained from World Bank) has been estimated. We assume the GDP at constant prices (USD 2005) to change at the similar rate as GDP-PPP in future. We obtain GDP at constant price (USD 2005) for the year 2010–2011 for all the states from the Reserve Bank of India Statistics (2014) and regrid it at $0.5° \times 0.5°$ resolution in GIS-based platform. For evaluating GDP (at constant USD 2005 prices) in future for the 5 SSPs scenarios for each decade in future, we apply the rate of change of GDP-PPP for that decade uniformly to each grid. Mathematically, it can be represented as in Eq. 7

$$GDP^{USD\,constant\,prices}_{decade=x} = GDP^{USD\,constant\,prices}_{year=2005}$$
$$+ \left[ \frac{\frac{GDP_{ppp\,decade=x,IIASA}^{SSP=w} + GDP_{ppp\,decade=x,OECD}^{SSP=w}}{2}}{GDP_{ppp\,year=2010}^{world\,bank}} \right.$$
$$\left. - GDP_{PPP\,year=2010}^{world\,bank} \right]$$
$$\times GDP^{USD\,constant\,prices}_{year=2005} \quad (7)$$

where $GDP^{SSP=w}_{decade=x}$ is the GDP for a given SSP scenario '$w$' and a future decade '$x$' and $GDP^{SSP=w}_{PPP\,decade=x,IIASA}$ is the GDP$_{PPP}$ projected by IIASA for India in each decade '$x$' in future for a SSP scenario '$w$'. Similarly, $GDP^{SSP=w}_{PPP\,decade=x,OECD}$ is the GDP$_{PPP}$ projected by OECD for each decade '$x$' for India in future for a SSP scenario '$w$'. The rate of change of the average GDP$_{PPP}$ (using IIASA and OECD) relative to $GDP^{world\,bank}_{PPP\,year=2010}$ (World Bank GDP$_{PPP}$ for India) is applied to GDP at constant prices for each Indian state, $GDP^{USD\,constant\,prices}_{year=2005}$ to obtain the GDP at constant prices for each future decade up to 2100. The state-wise estimates are then re-gridded at $0.5° \times 0.5°$ resolution for estimation of baseline mortality using the non-linear relations for the three diseases (COPD, IHD and stroke), while for LC it is assumed to remain constant throughout the century[10] as the LC baseline mortality does not show any relation with GDP.

The uncertainties in the estimates of premature mortality burden estimation stems from the disagreement in $PM_{2.5}$ estimated by the CMIP5 models and the uncertainty in estimation of baseline mortality from GDP using relations from our previous study[10]. The entire methodology is schematically explained in Supplementary Fig. 7.

**Underlying assumptions.** Premature mortality burden in India due to ambient $PM_{2.5}$ exposure is projected till the end of the century as discussed above based on the following underlying assumptions. First, the country-level age-specific fraction of population obtained from https://tntcat.iiasa.ac.at/SspDb/dsd?Action=htmlpage&page=about is applied uniformly for all the grids. Second, the present day GDP baseline mortality non-linear relation in India is assumed to hold true for future as well. Third, we consider that the IERs used to estimate RR holds true in the future decades. RR in the IERs depends only on the $PM_{2.5}$ mass concentration, not on the composition. Some recent studies have highlighted that RR for cardiovascular diseases due to $PM_{2.5}$ exposure dominated by biomass burning might not be the same as of $PM_{2.5}$ dominated by fossil-fuel combustion[61,62]. The IERs may be updated in future with inclusion of new cohort studies. Finally, we have not used emissions projected by the SSP scenarios to estimate future $PM_{2.5}$ exposure. Our estimated $PM_{2.5}$ exposure is driven by RCP scenarios only. We also have not considered air pollution control policies suggested in SSP scenarios[63] that may be implemented in India in future. Deviation from these assumptions would alter the estimated premature mortality burden and is a scope of future research.

**Code availability**. The codes used to arrive at the results depicted in this study are available upon request to the corresponding author (SC).

**Data availability**. Data for CMIP5 models are available at https://esgf-node.llnl.gov/projects/esgf-llnl/. All SSP data are available at https://tntcat.iiasa.ac.at/SspDb/dsd?Action=htmlpage&page=about. Gridded population data for all SSPs are available at http://clima-dods.ictp.it/Users/fcolon_g/ISI-MIP. MISR aerosol data are archived in NASA Langley Research Atmospheric science Data Center. All

other data as reproduced by this paper can be obtained upon request to the corresponding author (S.C.).

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

## Acknowledgements

We acknowledge financial support from the Department of Science and Technology, Govt. of India through a research grant (DST/CCP/(NET-2)/PR-36/2012(G)) under the network program of climate change and human health, which was operational at IIT Delhi (RP2726). We thank Lawrence Livermore National Laboratory (http://cmip-pcmdi.llnl.gov/cmip5/) for maintaining the CMIP model output datasets. We also thank the IIASA for maintaining the SSP database (https://tntcat.iiasa.ac.at/SspDb/dsd?Action=htmlpage&page=about). We thank the website http://clima-dods.ictp.it/Users/fcolon_g/ISI-MIP for maintaining the projected SSP gridded population data. We thank Aaron van Donkelaar (Dalhousie University, Halifax) for providing us the GEOS-Chem simulation of PM2.5/AOD ratio. We thank NASA Langley Research Atmospheric Science Data Center for archiving MISR aerosol products.

## Author contributions

S.C. and S.D. developed the idea, and S.C. carried out the analysis. S.D. and K.R.S. provided important inputs. All the authors wrote the paper.

## Additional information

**Competing interests:** The authors declare no competing financial interests.

