## [Peer Review File · Nature Communications]

Editorial Note: This manuscript has been previously reviewed at another journal that is not operating a transparent peer review scheme. This document only contains reviewer comments and rebuttal letters for versions considered at Nature Communications. Mentions of prior referee reports have been redacted.

Reviewers' comments:

Reviewer #1 (Remarks to the Author):

The manuscript "Ambient PM_{2.5} exposure and Expected Mortality in India under coupled climate change socio-economic scenarios" presents an extensive work on the human health impacts of air pollution through out the century in India. Such a vast and spatially detailed analysis focused on India adds novelty to the currently growing literature on air pollution impacts based on the SSPs and RCPs scenarios that has so far not yet looked at local scale mortality.

The authors find that PM_{2.5} exposure is bound to increase until the middle of the century in India and the effects of future climate on local meteorology mitigate around 10% of the rise in PM_{2.5} concentration in both the RCPs considered. The smallest burden is found in SSP4-RCP4.5, but those depend evidently on the future mortality rates and the population.

The idea behind the paper is very useful and important not only for the air pollution impacts community but also for the climate community. The new set of scenarios, SSPs, coupled with the already well known RCPs provides valuable insights on what will drive the air pollution impacts and what are the most crucial types of policies that are needed in the future for India. The authors also study the most important drivers of mortality due to PM_{2.5}. This paper is very important because it not only emphasizes the importance of air pollution policies but also bring light on how it relates with future climate-driven meteorology and a span of possible socio-economic futures.

Despite the idea behind the paper being very good, the manuscript itself is presented in a sloppy way. The figures are poorly organized, the figures and table labels and captions are not clear and descriptive enough, the units of metrics are not explicit. The methodology is not clear enough, and it is presented very late in the paper, and the scenario description is very confusing. It is not entirely understandable when and how the SSPs baselines are used.

General comments:

The authors use the term "coupled scenarios" throughout the manuscript and in the title. I am concerned about the interpretation the authors gave to the scenarios they have designed. The CMIP5 scenarios are based on the RCPs radiative forcing and assume only GHGs emissions/concentrations pathways; these have not been coupled with the SSPs. There is indeed very extensive work on coupled SSPs-RCPs scenarios published by the integrated assessment (IA) modelling community, such as in Riahi et al., 2016 <http://dx.doi.org/10.1016/j.gloenvcha.2016.05.009>. Here there is no such coupling, the authors assume that all combinations of SSPs-RCPs are possible, which is not feasible as proven in Riahi et al., 2016. The authors apply population and GDP SSP data on 2 RCPs, it is known that for example no IA model found the SSP1-RCP8.5 scenario feasible, additionally, the authors do not assume different emissions/concentrations pathways across SSPs, as far as I could understand, but only across RCPs. An SSP3-RCP4.5 yields a different PM_{2.5} concentration than a SSP1-RCP4.5 scenario. These concerns are reflected on sentences like "especially when the population distribution exposed to the pollution and baseline mortality are expected to change in response to changes in socio-economic condition to cope up with climate change impacts." There is no feedback on the population on the SSPs-RCPs, only GDP is changed through the carbon prices and investments on the energy system. This change should also be taken into account when projecting future

mortality. From what I have understood this is not the case. Again on sentence "...It can be argued that a given emission pathway defined by each of the RCP scenario may be reached under a wide variety of socioeconomic conditions" a given total radiative forcing may be achieved under different SSPs but it is very unlikely to achieve the same emission pathway under such a great variety of socioeconomic conditions such as the SSPs. The interpretation in sentence "These coupled scenarios represent the demographic change in response to climate change impacts." reflects the same concern that there is no feedback on the population in the SSPs or SSPs-RCPs scenarios. I suggest a clarification of on this.

1) The document needs a better organization, possibly adding sections and providing a methods / scenario matrix/table descriptive of the variable variations. Clearer information on resolution data origin and methods by scenario and analysis is needed.

2) The data here presented in the form of maps, would be more useful if it was disseminated at the state/grid level on an online public database in order to be assessed by the local policy makers and by the research community.

3) Something interesting can arise from the sensitivity analysis performed here, is that mortality or increase in mortality is solely not a good indicator of the impact of a given future scenario, different metrics can arise as, for instance, mortality per capita. The authors might use such metric to isolate the effects of a growing population better and better understand how the air pollution problem evolves throughout the century.

4) Another important aspect is the consideration of the air pollution policies in the SSPs; each SSP has an air pollution control policy storyline which is significant regarding air pollution exposure Rao et al. 2016 <http://dx.doi.org/10.1016/j.gloenvcha.2016.05.012> . One can not state that a given SSP yields better outcomes in terms of air pollution ignoring the effects of air pollution control policies on the exposure. I do understand that at this spatial detail level this does not exist as it is not considered in the CMIP5 database, however, the authors should be knowledgeable of this when referring to the results of a given SSP. They are not using the whole SSP story line but only a small part of the SSP database which does not reflect the storyline.

Specify comments:

1) The abstract should include a (or more) sentence on the methods(tool/database) of the PM2.5 projections.

2) The use of parenthesis to refer to two different results throughout the text is very confusing.

3) In the abstract, the authors state "Ambient PM2.5 concentration is expected to increase by 20% (28%) in 2030 (2050) due to rise in emissions", Figure 1 of the same document is not consistent with this result, unless this is across RCPs and also SSPs? Please clarify. Additionally is not compatible with the following sentence "PM2.5 is projected (following the first approach, see Methods for details) to increase till 2030 (2040) followed by a decrease till the end of the century under RCP4.5 (RCP8.5) scenario." In line 116.

4) The explanation on line 63 should come next to the statement in line 53-54.

5) Please provide an explanation for only using 2 of the RCPs, instead of all.

6) The authors mention the GDE, have the local authorities announced new measures after the episode? If yes, please refer to it.

7) In "Ambient PM2.5 Projection" the authors refer to "relative change in PM2.5", but they do not specify if this is concentrations or emissions, please clarify. The method in this section needs a better clear description, possibly using a mathematical equation. Additionally please specify if this is a new method or if it has been used before, and if yes please present a reference.

8) Figure S1 in the SI has no reference to the year or period; the dots represent a grid cell from which year? Please clarify. Please add the region to the map if you refer to them in the text such as Rajasthan and IGB.

9) In line 123 the ratio AOD/PM2.5 is specified differently in methods line 403 where it is the inverse.

10) The sentence in line 131-132 "while under the RCP8.5 scenario, increase in emission strength subdues the effect of meteorology for all the decades except the last one" is not consistent with figure 1, please explain or correct.

11) From line 151 to 156 it seems the authors are referring to the famous SSP diagram, from O'Neill, B.C., Kriegler, E., Riahi, K. et al. *Climatic Change* (2014) 122: 387. doi:10.1007/s10584-013-0905-2 which is not presented in this article, and not to figure 4.

12) It is not clear, in Figure 3 and the whole analysis, if emissions/concentration change with the SSPs or with the SSPs-RCPs. Please clarify this point when describing the scenarios and the results.

13) Please define burden and its unit.

14) Please harmonize the x-axis across the graphics; some refer to years some to periods. In figure one, please remove the axis ticks that have no label and do not represent a date since only periods are represented.

15) In figure 3 the shaded areas represent the model uncertainty range, but is this range related to the difference in population values or to the difference in model concentrations? Please explain.

16) Figure 1a and 1b are completely disconnected in the text. Please provide a better organization.

17) In figure 1b, it is not clear what is presented, what are the reference values and what the values and signs (positive and negative) mean. Reading the caption, it seems that the meteorology in RCP8.5 will aggravate the air pollution impact, however in the text we can read "...while projected steep increase in emission under RCP8.5 scenario subjugates meteorology.". If the change here is only due to meteorological conditions, why do the authors talk about the growing emissions? Meteorology should be isolated so that we can answer the question "despite the growing emissions does RCP induced meteorology alleviates or aggravates premature death from air pollution in RCP4.5 and RCP8.5?".

18) Rephrase sentence in line 193.

19) Please provide a scenario matrix or table; the results are all mixed and presented in a confusing order. Please provide a rationale for the relationship between mortality and GDP; mortality generally decreases with GDP.

20) In figure S3 what does the shaded area represent?

21) Please present table 2 in an intuitive way, with the explicit labels for the RCPs.

22) In line 205 table S4 is referred but it does not exist.

23) Please clarify what drives the differences between RCPs in Table 1 SA2, shouldn't PM2.5 exposure be the same for the baseline period? Are population and mortality(GDP) changing across RCPs?

24) The rationale provided for why under RCP4.5 for SA2 the sign is positive and for RCP8.5 is negative is confusing: "while for RCP4.5-SSP1, changes in socio-economic variables are large enough to over-shadow the effect of increasing PM2.5 exposure in modulating burden", isn't it that the concentrations from baseline are higher in baseline than for distant future in RCP4.5 and even the effects of the changing population and mortality cannot cover for this concentration difference?

25) Please clarify if it is considered that the exposed population is the total population or if it is a part of it and if this changes across the scenarios.

26) In lines 196-197, it is stated that "In the third sensitivity analysis (SA3), we keep baseline mortality and population to the levels of baseline period", and later it is stated referring to the SA3 that "the differences increase with time whereas the variation across SSP scenarios depends on the respective spatial variation in baseline mortality and population". Shouldn't these be the same by scenario definition? What is driving the differences in SA3 across SSPs? Please explain this is very confusing.

27) The last scenario description, scenarios (i) and (ii) is confusing, why isn't this part connected to the other SAs scenarios? And why is it presented with another format table?

28) Please provide the reference for CALIOP.

29) Please explain/provide the reference from where the value stated in line 426 is taken from.

30) The link provided in line 436 does not work.

31) Please provide a source for your baseline mortality data.

32) Please clarify the rationale presented in line 445-446, at the moment for me it is no understandable.

33) In Figure 2 please provide the labels (RCPs in the vertical axis and time periods on the horizontal axis), please provide a label for the color scale. Same in figure S2.

34) In Figure 4 provide the unit, and please redo the figure in an intuitive way, I suggest panels for RCPs (vertical) with the time in the x-axis, and colors for the SSPs with a color label legend.

Style and format comments:

Spell out all the acronyms on the first reference, such as RCPs, SSPs, AOD and IGB...

Harmonize the references either "Fig" or "Figure".

Please revise the manuscript in order to correct all the misspelled words and missing whitespaces both in the manuscript and in the SI, including captions.

Figure 4 is mentioned before figure 3.

Reviewer #2 (Remarks to the Author):

In this study, the authors project PM_{2.5} trends out to the year 2100 in conjunction with two climate change scenarios. They then apply concentration response functions to estimate the changes in premature mortality under five different socioeconomic scenarios. While the research idea would be of interest to the climate impacts, pollution exposure and health outcome fields, and satisfies the general interest requirement, the analysis should be strengthened for publication in this journal.

The manuscript is somewhat difficult to read and some figures are difficult to understand. For example, in Fig 4, it is not clear what the numbers signify. The figure axis is TDs' but the acronym is not defined. Also, the authors refer to an inset which does not exist. The numbers and uncertainties in the figures should be thoroughly explained and should have units. In addition to some of the more technical suggestion below, I urge the authors to take a very close look at the writing in the main text and figures before the next review.

One major concern with the analysis is the disconnection between the SSPs and RCPs. Since the changing socioeconomics do not influence the RCPs, I have an objection to the use of the word "coupled". Since SSPs and RCPs are independent, "coupled" is not the right term. More important, if RCPs and SSPs are mutually exclusive, the analysis may be flawed in a major way. The authors cite Vuuren et al. (2014) in arguing that RCPs may be reached under a wide variety of socioeconomic conditions but there must be certain limits to what is feasible. For example, can RCP8.5 scenario be reached under all the SSPs considered here, especially SSP1, which is the green growth scenario?

In regards to the suggested use of the analysis, SSPs are global strategies. Have their applicability to the conditions of India been considered? Further, the definition of "low mortality" or "high education" can vary significantly from country to country; therefore, their meaning for India should be discussed.

I also have a concern with the idea of using a constant AOD/PM_{2.5} ratio to estimate PM_{2.5} concentrations into the future. There are large uncertainties in the relations between AOD and ground-level PM_{2.5}. In general, these relations seem to be highly dependent on the composition of the PM_{2.5}. When the emission profiles change in the distant and far future, the present AOD/PM_{2.5} ratios may not be applicable because of the changes in the composition of PM_{2.5}. A better approach would be to project source specific emissions and estimate PM_{2.5} concentrations from those emissions. At this point, I want to note a general lack of uncertainty analysis in this study. The uncertainties associated with assumptions like this one should, at a minimum, be discussed.

Another concern is with the health findings, some of which may be farfetched. Given the shape of the standard concentration response function, where is India today so that "premature mortality burden is expected to increase by >100% by mid-of-the-century." With a landmass average of 35 micrograms, I would imagine that the baseline is already somewhere on the flatter portion of the concentration response curve. How realistic is it that an increase to 40-45 micrograms (Fig 1) would yield the suggested increases in the premature mortality burden?

Reviewer #3 (Remarks to the Author):

Major Comments

This analysis includes what would seem to be some new and potentially useful information given

its focus on India and the linkage between air pollution-related mortality and future climate and socioeconomic scenarios, in its present form it is hard to see how the results and analyses would be useful or meet the authors stated aim of being useful in “formulating policy that is co-beneficial to climate change and air quality.” Without more emphasis on the key components of the different scenarios and how they impact the results, the analysis is little more than a description of future projections under a mix of different hypothetical scenarios.

While I appreciate the ~85 year time horizon with respect to global temperature change, the uncertainty with regard to mortality (especially in a rapidly developing economy such as India) over this period is huge. Indeed, there is very little variation between the different RCP or SSP scenarios until after 2050, but beyond this time point demographic and epidemiologic projections are highly uncertain. Perhaps instead of emphasizing what seem to be large projected differences between 2050 and 2100 it would be more appropriate to emphasize that the analyses suggest essentially no difference in projected mortality until 2050 under the different scenarios. I question however, whether this a realistic finding, or how one should interpret an analysis that would appear to assume high certainty regarding future air pollution-related mortality in India over the next 35 years.

How likely is it that there will be NO policy interventions by the Government with regard to air pollution in the future? This seems highly unrealistic and at least should be considered in sensitivity analyses. In fact, the authors state that they expect the government to implement policy interventions. The inclusion of sensitivity analysis with regard to meteorology but not policy interventions seems misguided. Are the authors suggesting that meteorologic changes related to climate warming are likely to be more important for future air pollution-related mortality than policy interventions?

The authors assume uniform baseline mortality which is entirely unrealistic over the timeframe of analysis. Instead the sensitivity analysis of adjusted mortality should be in the core of the analysis. Further, although included in the sensitivity analyses, the core analyses do not seem to account for an aging population. Given that most of the causes of death affected by air pollution impact the older segments of the population (COPD, Stroke, IHD, Lung Cancer) as well as lower respiratory infections which affect the very young and the very old, the baseline mortality is highly sensitive to the age structure of the population. With increasing socioeconomic development in the future India's population (which is now relatively young) will age considerably. More generally, the manuscript would strongly benefit from a succinct presentation of sensitivities in the near-term and long term to a)emissions b)meteorology c)population growth d)baseline mortality e)population aging

L178 While there is some discussion of meteorologic impacts which would decrease future PM2.5 and attributable mortality would there be no increase in secondary particulate matter production? The latter is not mentioned at all in the text. Further, most warming scenarios suggest increased substantially increased ozone concentrations, to the degree that (global) ozone-related mortality may surpass that related to PM in the latter part of the century. There is no mention of ozone in the analysis which seems like a major shortcoming given the time horizons that are included.

What is the basis for the assumption of a GDP-mortality relationship for India? This could (and should) be evaluated based on historical data

Overall it is very hard to follow the Methods. The manuscript would benefit from a schematic describing the different inputs and how they change for the different analyses. As well, I would suggest that a more limited set of scenarios (either divergent scenarios to bound the projections or the most likely combinations) and assumptions could be the focus for the main text with other scenarios/assumptions provided in the Supplementary Material. At present it is hard to really pull out the key findings.

Specific Comments

Abstracts typically do not include references

Should either provide a range of attributable mortality estimates for India with specific citations (and discussion of differences) or use the most recent one (Global Burden of Disease 2015)

Abstract refers to "Great Diwali Air Pollution Episode of Nov 2016" which is neither especially unusual for Delhi nor likely to be well known to readers – suggest removing reference in abstract and main text.

Abstract contains abbreviations that are not described – e.g. SS3

Manuscript needs to be thoroughly edited for grammar and jargon – e.g. "India, second-most populous country in the world, is recognized as a hotbed for aerosol loading"

The Discussion would benefit from some mention of how warming may/may not affect baseline mortality in India.

For non-experts some brief description of the different RCP scenarios should be provided.

Reviewers' comments:

Reviewer #1 (Remarks to the Author):

The manuscript “Ambient PM2.5 exposure and Expected Mortality in India under coupled climate change socio-economic scenarios” presents an extensive work on the human health impacts of air pollution through out the century in India. Such a vast and spatially detailed analysis focused on India adds novelty to the currently growing literature on air pollution impacts based on the SSPs and RCPs scenarios that has so far not yet looked at local scale mortality.

The authors find that PM2.5 exposure is bound to increase until the middle of the century in India and the effects of future climate on local meteorology mitigate around 10% of the rise in PM2.5 concentration in both the RCPs considered. The smallest burden is found in SSP4-RCP4.5, but those depend evidently on the future mortality rates and the population.

The idea behind the paper is very useful and important not only for the air pollution impacts community but also for the climate community. The new set of scenarios, SSPs, coupled with the already well known RCPs provides valuable insights on what will drive the air pollution impacts and what are the most crucial types of policies that are needed in the future for India. The authors also study the most important drivers of mortality due to PM2.5. This paper is very important because it not only emphasizes the importance of air pollution policies but also bring light on how it relates with future climate-driven meteorology and a span of possible socio-economic futures.

Despite the idea behind the paper being very good, the manuscript itself is presented in a sloppy way. The figures are poorly organized, the figures and table labels and captions are not clear and descriptive enough, the units of metrics are not explicit. The methodology is not clear enough, and it is presented very late in the paper, and the scenario description is very confusing. It is not entirely understandable when and how the SSPs baselines are used.

We thank the reviewers for appreciating the idea behind the manuscript. We admit that the previous version was not very well organized. We have attempted to improve the manuscript by modifying the figures, the table labels and captions. We have modified the methodology to make it more lucid and readable. We have used the populations that were used as drivers to develop the SSP scenarios and RCP4.5 and RCP8.5 concentration to estimate the premature mortality.

General comments:

The authors use the term “coupled scenarios” throughout the manuscript and in the title. I am concerned about the interpretation the authors gave to the scenarios they have designed. The CMIP5 scenarios are based on the RCPs radiative forcing and assume only GHGs emissions/concentrations pathways; these have not been coupled with the SSPs. There is indeed very extensive work on coupled SSPs-RCPs scenarios published by the integrated assessment (IA) modelling community, such as in Riahi et al., 2016 <http://dx.doi.org/10.1016/j.gloenvcha.2016.05.009> . Here there is no such coupling, the authors assume that all combinations of SSPs-RCPs are possible, which is not feasible as proven in Riahi et al., 2016. The authors apply population and GDP SSP data on 2 RCPs, it is known that for example no IA model found the SSP1-RCP8.5 scenario feasible, additionally, the authors do not assume different emissions/concentrations pathways across SSPs, as far as I could understand, but only

across RCPs. An SSP3-RCP4.5 yields a different PM2.5 concentration than a SSP1-RCP4.5 scenario. These concerns are reflected on sentences like “especially when the population distribution exposed to the pollution and baseline mortality are expected to change in response to changes in socio-economic condition to cope up with climate change impacts.” There is no feedback on the population on the SSPs-RCPs, only GDP is changed through the carbon prices and investments on the energy system. This change should also be taken into account when projecting future mortality. From what I have understood this is not the case. Again on sentence “...It can be argued that a given emission pathway defined by each of the RCP scenario may be reached under a wide variety of socioeconomic conditions” a given total radiative forcing may be achieved under different SSPs but it is very unlikely to achieve the same emission pathway under such a great variety of socioeconomic conditions such as the SSPs. The interpretation in sentence “These coupled scenarios represent the demographic change in response to climate change impacts.” reflects the same concern that there is no feedback on the population in the SSPs or SSPs-RCPs scenarios. I suggest a clarification of on this.

We thank the reviewer for such comprehensive review of our article. We feel that majority of the comments arise due to the lack of clarity in our previous version. We have modified the manuscript with elaborate description of the various assumption and scenarios considered in this study. We acknowledge the detailed studies done on coupled RCP-SSP scenarios. We clarify that in this work we use socio-economic factors like population and GDP which were used as drivers in developing SSP scenarios; and not the emissions (based on which concentration can be calculated) projected by SSP scenarios. We estimate the concentration from CMIP5 models under 2 RCP scenarios. We use concentrations of PM2.5 as projected for RCP scenarios. The discussion of the results are modified accordingly. We acknowledge from Riahi et al, 2017

<http://dx.doi.org/10.1016/j.gloenvcha.2016.05.009> that RCP8.5 and SSP1 combination is impossible and have slashed out the RCP8.5 –SSP1 estimates throughout the manuscript.

We have now revamped the sensitivity analysis completely, SA1 now indicates change of burden of premature mortality/year due to demographic transition, SA2 represents change of burden of premature mortality/year due to epidemiological transition, SA3 represents change of premature mortality/year due to meteorological change and SA4 represents change of premature mortality per year due to implementation of certain policy measures. The detailed definition of the sensitivity studies are illustrated in a new table (Table 1) in the revised manuscript.

1) The document needs a better organization, possibly adding sections and providing a methods / scenario matrix/table descriptive of the variable variations. Clearer information on resolution data origin and methods by scenario and analysis is needed.

Thank you for suggesting this, we attempted to organize the sections. A flowchart describing the methods is provided in the supplementary material. Supplementary material Figure S7.

2) The data here presented in the form of maps, would be more useful if it was disseminated at the state/grid level on an online public database in order to be assessed by the local policy makers and by the research community.

We thank the reviewers this suggestion. We have now added a table in the Supplementary material (Supplementary material Table S3) containing burden of premature mortality at state level.

3) Something interesting can arise from the sensitivity analysis performed here, is that mortality

or increase in mortality is solely not a good indicator of the impact of a given future scenario, different metrics can arise as, for instance, mortality per capita. The authors might use such metric to isolate the effects of a growing population better and better understand how the air pollution problem evolves throughout the century.

We thank the reviewer for this comment. We have now added a new table as Supplementary material Table S4 which depicts premature mortality per capita (per 100000 population). We have also added this in discussions under section 'Projection of premature death due to ambient PM_{2.5} exposure' in line numbers 172-176, 183-188 and 188-190

4) Another important aspect is the consideration of the air pollution policies in the SSPs; each SSP has an air pollution control policy storyline which is significant regarding air pollution exposure Rao et al. 2016 <http://dx.doi.org/10.1016/j.gloenvcha.2016.05.012> . One can not state that a given SSP yields better outcomes in terms of air pollution ignoring the effects of air pollution control policies on the exposure. I do understand that at this spatial detail level this does not exist as it is not considered in the CMIP5 database, however, the authors should be knowledgeable of this when referring to the results of a given SSP. They are not using the whole SSP story line but only a small part of the SSP database which does not reflect the storyline.

We thank the reviewer for informing this to us. This can be an entire new work. We acknowledge that we do not take into consideration the entire SSP story line but only a part of it. We do not take into consideration the SSP air control policies in future on our estimates of premature mortality burden. We mentioned this in the section 'underlying assumptions' (Lines 435-436)

Specify comments:

1) The abstract should include a (or more) sentence on the methods(tool/database) of the PM2.5 projections.

We thank the reviewer for this comment. We have added a couple of sentences on the method of PM_{2.5} projection (Lines 31-34)

2) The use of parenthesis to refer to two different results throughout the text is very confusing.

We acknowledge that the representation was confusing and have made it simpler. We no further use parenthesis to refer to two different cases.

3) In the abstract, the authors state "Ambient PM2.5 concentration is expected to increase by 20% (28%) in 2030 (2050) due to rise in emissions", Figure 1 of the same document is not consistent with this result, unless this is across RCPs and also SSPs? Please clarify. Additionally is not compatible with the following sentence "PM2.5 is projected (following the first approach, see Methods for details) to increase till 2030 (2040) followed by a decrease till the end of the century under RCP4.5 (RCP8.5) scenario." In line 116.

We thank the reviewer for pointing this out. It has been revised now. (Lines 34-36)

4) The explanation on line 63 should come next to the statement in line 53-54.

It has been changed now.

5) Please provide an explanation for only using 2 of the RCPs, instead of all.

We thank the reviewer for the comment. We have now provided explanations for using only two RCP scenarios instead of all. (Lines 81-83)

6) The authors mention the GDE, have the local authorities announced new measures after the episode? If yes, please refer to it.

After the episode, graded action plan has been implemented, but restricted to Delhi only. It is too short to assess its success. At the moment, no such measures are taken anywhere else in the country. We anticipate that in future such measures may be taken. Hence we added one sensitivity study (SA4) where we assume that in near future all-India average PM_{2.5} meets WHO IT-1 due to the policy intervention at larger spatial scale. The implications are discussed in the revised manuscript. Although we have removed all references for GDE from the main paper as one of the other reviewers demanded it. Though we mention it briefly in the 'Concluding remarks' section. (Lines 286-294).

7) In "Ambient PM_{2.5} Projection" the authors refer to "relative change in PM_{2.5}", but they do not specify if this is concentrations or emissions, please clarify. The method in this section needs a better clear description, possibly using a mathematical equation. Additionally please specify if this is a new method or if it has been used before, and if yes please present a reference.

We thank the reviewer for this comment, we add up the components of PM_{2.5} obtained from 13 CMIP5 models to obtain the concentration of PM_{2.5}. We have now further clarified it in the 'Methods' section and in the section 'Ambient PM_{2.5} projection' (Lines 363-369 and lines 107-112)

This method to correct for the underestimation in CMIP5 models using satellite derived PM_{2.5} is new. We have mentioned it now in line 365-366.

We have now illustrated the methodology as a flowchart in Supplementary material Figure S7.

8) Figure S1 in the SI has no reference to the year or period; the dots represent a grid cell from which year? Please clarify. Please add the region to the map if you refer to them in the text such as Rajasthan and IGB.

Thank you for pointing this out. Now we have updated the figure caption of supplementary material Figure S1 with reference to the period (2011-2015). Each dot represents average exposure for the period of interest (mentioned in figure caption in the Supplementary material Figure S1. Rajasthan is now marked with a black box in supplementary material Figure S1. It is also mentioned in main text (Lines 98-107)

9) In line 123 the ratio AOD/PM_{2.5} is specified differently in methods line 403 where it is the inverse.

Thank you for pointing it out. The ratio or the conversion factor is expressed as PM_{2.5}/AOD. It has now been corrected in Line 118.

10) The sentence in line 131-132 "while under the RCP8.5 scenario, increase in emission strength subdues the effect of meteorology for all the decades except the last one" is not consistent with figure 1, please explain or correct.

We thank the reviewer for this comment. We have corrected the explanation accordingly in lines 128-132.

11) From line 151 to 156 it seems the authors are referring to the famous SSP diagram, from O'Neill, B.C., Kriegler, E., Riahi, K. et al. Climatic Change (2014) 122: 387. doi:10.1007/s10584-013-0905-2 which is not presented in this article, and not to figure 4.

We thank the reviewer for pointing this out We agree that we were attempting to refer to the famous SSP diagram from O'Neill et al., 2014. We have now put the figure in the Supplementary material Figure S3. It is also mentioned in main text line 151.

12) It is not clear, in Figure 3 and the whole analysis, if emissions/concentration change with the SSPs or with the SSPs-RCPs. Please clarify this point when describing the scenarios and the results. We thank the reviewer for pointing this out. We clarify that we use concentration estimated from the CMIP5 models. Population and socioeconomic projections which were used as drivers for the SSP scenarios were combined with the RCP concentration. Concentration does not change for the combination of a RCP and the 5 SSPs'. For example, concentration does not change for the combinations RCP4.5-SSP4 and RCP4.5-SSP5. But concentration changes between the combinations RCP4.5-SSP4 and RCP8.5-SSP4. It has been clarified in Lines 156-162

13) Please define burden and its unit.

Burden refers to premature mortality burden, its unit is premature mortality/year. The unit is now mentioned in the subsection heading (Line No. 144)

The definition is provided in' Methods' section (Lines 386-396)

14) Please harmonize the x-axis across the graphics; some refer to years some to periods. In figure one, please remove the axis ticks that have no label and do not represent a date since only periods are represented.

We thank the reviewers for pointing this out. We have harmonized the axes for the figures across the manuscript.

15) In figure 3 the shaded areas represent the model uncertainty range, but is this range related to the difference in population values or to the difference in model concentrations? Please explain.

The upper limit of the range signifies the premature mortality estimated using the (mean+ σ) $PM_{2.5}$ exposure (where σ is the standard deviation) for each grid obtained from 13CMIP5 models (signifies the spread across models) and the (central value + error) values of the coefficients that were used in the formulation of the relation between GDP and Baseline mortality. The lower limit of the range signifies the premature mortality estimated using (mean- σ) $PM_{2.5}$ exposure for each grid obtained from 13CMIP5 models and the (central value - error) values of the coefficients that were used in the formulation of the relation between GDP and baseline mortality.. It has been explained in lines 202-209

16) Figure 1a and 1b are completely disconnected in the text. Please provide a better organization.

We thank the reviewer for this comment. We have now removed Figure 1b. We have put the change in premature mortality as numbers in Table2.

17) In figure 1b, it is not clear what is presented, what are the reference values and what the values and signs (positive and negative) mean. Reading the caption, it seems that the meteorology in RCP8.5 will aggravate the air pollution impact, however in the text we can read "...while projected steep increase in emission under RCP8.5 scenario subjugates meteorology.". If the change here is only due to meteorological conditions, why do the authors talk about the growing emissions? Meteorology should be isolated so that we can answer the question "despite the growing emissions does RCP induced meteorology alleviates or aggravates premature death from air pollution in RCP4.5 and RCP8.5?"

We thank the reviewer for raising this concern. We have removed Figure 1b. Which we understand was disconnected in context. We now name the sensitivity analysis for estimating the change in premature mortality burden due to change in meteorological factors in future as SA3. We agree that meteorology under RCP8.5 scenario will worsen the health burden after 2050 until the last decade of the century. It is explained in details now in Lines 248-261, and the numbers are illustrated in Table 1 and Table 2

18) Rephrase sentence in line 193.

We have removed the sensitivity study which demanded the sentence

19) Please provide a scenario matrix or table; the results are all mixed and presented in a confusing order. Please provide a rationale for the relationship between mortality and GDP; mortality generally decreases with GDP.

Scenario matrix is provided in Figure 4. It represents the total country level premature mortality/year for 3 representative decades. The Table (Table 1) illustrates the sensitivity analyses and the number of premature mortality that is averted/ aggravated due to various factors we considered. The relationship between GDP and baseline mortality is defined in our previous paper (<http://www.sciencedirect.com/science/article/pii/S0160412016300848>)

20) In figure S3 what does the shaded area represent?

Supplementary material Figure S3 is now numbered Supplementary material Figure S4. The shaded area represents the range of baseline mortality estimated from the range in the coefficients of the equation that relates GDP and baseline mortality. We have mentioned the range in the figure caption of Supplementary material Figure S4

21) Please present table 2 in an intuitive way, with the explicit labels for the RCPs.

We presume the reviewer is indicating Supplementary material Table S2, we have modified the table accordingly.

22) In line 205 table S4 is referred but it does not exist.

We removed the section which deserved this explanation. As mentioned earlier we have revamped the sensitivity analyses.

23) Please clarify what drives the differences between RCPs in Table 1 SA2, shouldn't PM2.5 exposure be the same for the baseline period? Are population and mortality(GDP) changing across RCPs?

We removed the section which deserved this explanation.

24) The rationale provided for why under RCP4.5 for SA2 the sign is positive and for RCP8.5 is negative is confusing: “while for RCP4.5-SSP1, changes in socio-economic variables are large enough to over-shadow the effect of increasing PM2.5 exposure in modulating burden”, isn’t it that the concentrations from baseline are higher in baseline than for distant future in RCP4.5 and even the effects of the changing population and mortality cannot cover for this concentration difference?

We removed the section which deserved this explanation

25) Please clarify if it is considered that the exposed population is the total population or if it is a part of it and if this changes across the scenarios.

The exposed population is the population (>25 years) for the first year of the decade, for example, if we are estimating premature mortality burden/year for 2031-40, population >25 years for the year 2031 is considered as the exposed population as we expect it to be the minimum population that will get chronically exposed to PM_{2.5} for that decade. For IHD and stroke we consider age specific population at 5 years interval from 25 to >80 to estimate age specific premature mortality. Population changes across the SSPs. For example RCP4.5-SSP4 is considered to have same population as of RCP8.5-SSP4. Whereas population is considered to vary between RCP4.5-SSP4 and RCP4.5-SSP5.

26) In lines 196-197, it is stated that “In the third sensitivity analysis (SA3), we keep baseline mortality and population to the levels of baseline period”, and later it is stated referring to the SA3 that “the differences increase with time whereas the variation across SSP scenarios depends on the respective spatial variation in baseline mortality and population”. Shouldn’t these be the same by scenario definition? What is driving the differences in SA3 across SSPs? Please explain this is very confusing.

This sensitivity study has now been removed. SA3 is now the change in premature mortality/year due to meteorological changes in the future.

27) The last scenario description, scenarios (i) and (ii) is confusing, why isn’t this part connected to the other SAs scenarios? And why is it presented with another format table?

We thank the reviewer for this suggestion, We have now made this part of SA, demographical transition is now SA1 and epidemiological transition is SA2 (see Table1 for SA definitions). The numbers are presented in Table 2. The description is now presented in Lines 222-247.

28) Please provide the reference for CALIOP.

We thank the reviewer for the suggestion. The reference is added in Line 330.

29) Please explain/provide the reference from where the value stated in line 426 is taken from.

Reference is added in line 387.

30) The link provided in line 436 does not work.

The link is changed now and is working now.

31) Please provide a source for your baseline mortality data.

Baseline mortality data has been generated by us in our previous work (<http://www.sciencedirect.com/science/article/pii/S0160412016300848>). See reference no. 8.

32) Please clarify the rationale presented in line 445-446, at the moment for me it is no understandable.

We assume that the population in the first year of a decade is the minimum population that gets exposed to the decadal $PM_{2.5}$ for that particular decade. The same rationale was used by us in our previous published paper (<http://www.sciencedirect.com/science/article/pii/S0160412016300848>)

33) In Figure 2 please provide the labels (RCPs in the vertical axis and time periods on the horizontal axis), please provide a label for the color scale. Same in figure S2.

We thank the reviewer for this suggestion. We have modified the figures accordingly

34) In Figure 4 provide the unit, and please redo the figure in an intuitive way, I suggest panels for RCPs (vertical) with the time in the x-axis, and colors for the SSPs with a color label legend.

We thank the reviewer for the comment. We have modified the figure accordingly.

Style and format comments:

Spell out all the acronyms on the first reference, such as RCPs, SSPs, AOD and IGB...

We have done so.

Harmonize the references either "Fig" or "Figure".

We have titled Figure for referring to the figures and Supplementary material Figure for referring the figures in Supplementary material.

Please revise the manuscript in order to correct all the misspelled words and missing whitespaces both in the manuscript and in the SI, including captions.

We have gone through the manuscript thoroughly to correct any misspelled words and remove white spaces

Figure 4 is mentioned before figure 3.

We have corrected for it now

Reviewer #2 (Remarks to the Author):

In this study, the authors project PM_{2.5} trends out to the year 2100 in conjunction with two climate change scenario. They then apply concentration response functions to estimate the changes in premature mortality under five different socioeconomic scenarios. While the research idea would be of interest to the climate impacts, pollution exposure and health outcome fields, and satisfies the general interest requirement, the analysis should be strengthened for publication in this journal.

The manuscript is somewhat difficult to read and some figures are difficult to understand. For example, in Fig 4, it is not clear what the numbers signify. The figure axis is TDs' but the acronym is not defined. Also, the authors refer to an inset which does not exist. The numbers and uncertainties in the figures should be thoroughly explained and should have units. In addition to some of the more technical suggestion below, I urge the authors to take a very close look at the writing in the main text and figures before the next review.

We thank the reviewer for such detailed comments. We acknowledge that most of the issues arise from the lack of clarity in the previous version of the manuscript. We have modified Figure 4. The numbers in figure 4 represent premature mortality burden/year. We have now added a new section 'Underlying assumptions' in Methods, that lists all the assumptions we have taken into consideration in this work. We have discussed the uncertainties that are depicted as shaded region in this paper. (Lines 113-114, 166-167, 203-210)

One major concern with the analysis is the disconnection between the SSPs and RCPs. Since the changing socioeconomics do not influence the RCPs, I have an objection to the use of the word "coupled". Since SSPs and RCPs are independent, "coupled" is not the right term. More important, if RCPs and SSPs are mutually exclusive, the analysis may be flawed in a major way. The authors cite Vuuren et al. (2014) in arguing that RCPs may be reached under a wide variety of socioeconomic conditions but there must be certain limits to what is feasible. For example, can RCP8.5 scenario be reached under all the SSPs considered here, especially SSP1, which is the green growth scenario?

We thank the reviewer for raising this concern. This issue was raised due to lack of clarity in our previous version of the manuscript. We would like to clarify that we estimate the PM_{2.5} exposure employing Approach 1, from the 13 CMIP5 models and use it along with projected population and socioeconomic factors, which were used as drivers for the SSP scenarios. These population projections are also known as reference scenarios as they do not consider any change in climate policies. We also acknowledge for example that RCP4.5-SSP4 may have different concentration than RCP4.5-SSP5 due to climate policies in different SSP scenarios, but in this work we have not considered this change, we use only a part of the SSP storyline which is the projected population which were used as drivers for the SSP scenarios. We have modified the manuscript throughout accordingly. We also acknowledge that all the combinations of SSP and RCP are impossible. Following from Riahi et al, 2017 <http://dx.doi.org/10.1016/j.gloenvcha.2016.05.009> we have slashed out the RCP8.5-SSP1 combination from all the tables and figures. (Line 150)

In regards to the suggested use of the analysis, SSPs are global strategies. Have their applicability to the conditions of India been considered? Further, the definition of "low mortality" or "high

education” can vary significantly from country to country; therefore, their meaning for India should be discussed.

We acknowledge the concern of the reviewer, we have considered the storyline for describing population of India for 5 SSP scenarios from a recent paper (<http://www.sciencedirect.com/science/article/pii/S0959378014001095>.) We have cited this paper. Please see line 33-37 in Supplementary material

I also have a concern with the idea of using a constant AOD/PM2.5 ratio to estimate PM2.5 concentrations into the future. There are large uncertainties in the relations between AOD and ground-level PM2.5. In general, these relations seem to be highly dependent on the composition of the PM2.5. When the emission profiles change in the distant and far future, the present AOD/PM2.5 ratios may not be applicable because of the changes in the composition of PM2.5. A better approach would be to project source specific emissions and estimate PM2.5 concentrations from those emissions. At this point, I want to note a general lack of uncertainty analysis in this study. The uncertainties associated with assumptions like this one should, at a minimum, be discussed.

We thank the reviewer for raising this issue. We would like to clarify that we estimate PM_{2.5} using Approach 2 to design a sensitivity analysis (SA3) that separates out the effect of meteorology in modulating premature mortality burden/year. Throughout the paper we estimate PM_{2.5} using Approach 1 in which we add the concentration of the PM_{2.5} components like black carbon, organic aerosols, sulphate, NH₄⁺ ion, a fraction of dust and sea salt.

Another concern is with the health findings, some of which may be farfetched. Given the shape of the standard concentration response function, where is India today so that “premature mortality burden is expected to increase by >100% by mid-of-the-century.” With a landmass average of 35 micrograms, I would imagine that the baseline is already somewhere on the flatter portion of the concentration response curve. How realistic is it that an increase to 40-45 micrograms (Fig 1) would yield the suggested increases in the premature mortality burden?

We thank the reviewer for this comment. We would like to clarify that 35µg/m³ falls in the steep part of the curve as in Burnett et al., 2014. We have now removed the Figure 3 (which considered constant baseline mortality as of present day along with changing PM_{2.5} and exposed population) as in the previous manuscript and replaced it with the figure that depicts analysis using changing baseline mortality as a function of GDP (This was part of the sensitivity analysis previously, but inducted as the main analysis as suggested by another reviewer). We would like to anyway clarify that premature mortality burden is a function of relative risk (which depends on concentration), baseline mortality (estimated as a function of GDP) and exposed population. Due to huge projected rise of GDP in future, Baseline mortality is expected to decrease significantly which results in drop of premature mortality in future even though population and PM_{2.5} are expected to increase until upto 2050.

Reviewer #3 (Remarks to the Author):

Major Comments

This analysis includes what would seem to be some new and potentially useful information given its focus on India and the linkage between air pollution-related mortality and future climate and socioeconomic scenarios, in its present form it is hard to see how the results and analyses would be useful or meet the authors stated aim of being useful in “formulating policy that is co-beneficial to climate change and air quality.” Without more emphasis on the key components of the different scenarios and how they impact the results, the analysis is little more than a description of future projections under a mix of different hypothetical scenarios.

We have tried to emphasize the impact of different key variables on the outcomes to give policy makers an understanding of possible futures in India.

While I appreciate the ~85 year time horizon with respect to global temperature change, the uncertainty with regard to mortality (especially in a rapidly developing economy such as India) over this period is huge. Indeed, there is very little variation between the different RCP or SSP scenarios until after 2050, but beyond this time point demographic and epidemiologic projections are highly uncertain. Perhaps instead of emphasizing what seem to be large projected differences between 2050 and 2100 it would be more appropriate to emphasize that the analyses suggest essentially no difference in projected mortality until 2050 under the different scenarios. I question however, whether this a realistic finding, or how one should interpret an analysis that would appear to assume high certainty regarding future air pollution-related mortality in India over the next 35 years.

This result is actually in accord with climate scenarios that show relatively small changes over the same period even though policies are different and the long-term outcomes diverge. This is one of the policy dilemmas of climate change.

How likely is it that there will be NO policy interventions by the Government with regard to air pollution in the future? This seems highly unrealistic and at least should be considered in sensitivity analyses. In fact, the authors state that they expect the government to implement policy interventions. The inclusion of sensitivity analysis with regard to meteorology but not policy interventions seems misguided. Are the authors suggesting that meteorologic changes related to climate warming are likely to be more important for future air pollution-related mortality than policy interventions?

We thank the reviewer for the detailed comment and constructive suggestion. We formulate a sensitivity analysis (SA4) that estimates the averted premature mortality if certain policy interventions are implemented. We assume that by the end of the decade 2031-40, all India average will meet $35 \mu\text{g}/\text{m}^3$ (WHO-IT1) due to policy interventions. Further we assume that by the end of the decade 2061-70 all India average meets WHO-IT2 ($25 \mu\text{g}/\text{m}^3$). We assume that by the last decade very stringent policy interventions may be implemented such that all India average $\text{PM}_{2.5}$ exposure meets $15 \mu\text{g}/\text{m}^3$ WHO-IT3). See line 263-281.

The authors assume uniform baseline mortality which is entirely unrealistic over the timeframe of analysis. Instead the sensitivity analysis of adjusted mortality should be in the core of the analysis. Further, although included in the sensitivity analyses, the core analyses do not seem to account for an aging population. Given that most of the causes of death affected by air pollution impact

the older segments of the population (COPD, Stroke, IHD, Lung Cancer) as well as lower respiratory infections which affect the very young and the very old, the baseline mortality is highly sensitive to the age structure of the population. With increasing socioeconomic development in the future India's population (which is now relatively young) will age considerably. More generally, the manuscript would strongly benefit from a succinct presentation of sensitivities in the near-term and long term to a) emissions b) meteorology c) population growth d) baseline mortality e) population aging

We sincerely thank the reviewer his suggestions. We have now restructured the paper completely. We put estimations of premature mortality burden/year using changing baseline mortality in the main paper. Lines 146-210.

We have also now revamped the sensitivity analysis completely, SA1 now indicates change of burden of premature mortality/year due demographic transition, SA2 represents change of burden of premature mortality/year epidemiological transition, SA3 represents change of premature mortality/year due to meteorological change. SA4 represents change of premature mortality per year due to implementation of certain policy measures. The detailed definition of the sensitivity studies are illustrated in a new table (Table 1) Lines 213-281.

We find that aged population is projected to increase in future under all the SSP scenarios (Supplementary material Figure S6). But it is difficult to formulate a sensitivity analysis on the aging population because the RR obtained from IER function are age specific for only IHD and stroke. So we need to wait until the GBD group publishes age specific IER for the other diseases It may be a good analysis for some future work.

L178 While there is some discussion of meteorologic impacts which would decrease future PM2.5 and attributable mortality would there be no increase in secondary particulate matter production? The latter is not mentioned at all in the text. Further, most warming scenarios suggest increased substantially increased ozone concentrations, to the degree that (global) ozone-related mortality may surpass that related to PM in the latter part of the century. There is no mention of ozone in the analysis which seems like a major shortcoming given the time horizons that are included.

We thank the reviewers for this comment. We completely agree with the reviewer, This is an innovative idea for a future work. In this paper we attempted to focus on future health effect of PM_{2.5}, so we chose not to include ozone, we have mentioned this briefly now in Lines 288-291.

What is the basis for the assumption of a GDP-mortality relationship for India? This could (and should) be evaluated based on historical data

We have published our previous paper, <http://www.sciencedirect.com/science/article/pii/S0160412016300848> in which we establish the relation between GDP and baseline mortality and have validated it for India in the same paper.

Overall it is very hard to follow the Methods. The manuscript would benefit from a schematic describing the different inputs and how they change for the different analyses. As well, I would suggest that a more limited set of scenarios (either divergent scenarios to bound the projections or the most likely combinations) and assumptions could be the focus for the main text with other scenarios/assumptions provided in the Supplementary Material. At present it is hard to really pull

out the key findings.

We thank the reviewer for this comment. We have now revamped the manuscript completely with lesser number of sensitivity analyses. We have also added Supplementary material Figure S7 which describes the methodology in a flowchart. We have added Table1 which schematically describes which inputs change in what sensitivity analysis.

Specific Comments

Abstracts typically do not include references

We have excluded reference from the abstract now.

Should either provide a range of attributable mortality estimates for India with specific citations (and discussion of differences) or use the most recent one (Global Burden of Disease 2015)

We have now mentioned the attributable mortality from GBD with reference (Line 51)

Abstract refers to “Great Diwali Air Pollution Episode of Nov 2016” which is neither especially unusual for Delhi nor likely to be well known to readers – suggest removing reference in abstract and main text.

We thank the reviewer for this comment. We have removed GDE from the abstract and main text although we prefer to discuss about it in the conclusion section

Abstract contains abbreviations that are not described – e.g. SS3

We have now spelled out the abbreviations in the abstract.

Manuscript needs to be thoroughly edited for grammar and jargon – e.g. “India, second-most populous country in the world, is recognized as a hotbed for aerosol loading”

We have attempted to edit for the grammatical jargons.

The Discussion would benefit from some mention of how warming may/may not affect baseline mortality in India.

We have added a discussion that includes some details on this issue. See line 263 and afterwards.

For non-experts some brief description of the different RCP scenarios should be provided

We have added a section in Supplementary material describing the RCP scenarios In brief. Supp. Material Line 65-91.

Reviewers' comments:

Reviewer #1 (Remarks to the Author):

I would like to congratulate the authors for the better organization, for the clearer methodology and for the new sensitivity analysis.

However I still think this paper is still not clear enough, I think a mathematical description of the methodology will help the rigour that is lacking. The paper still needs better wording, some sentences seem to be detached from context and above all it needs more rigour, some concepts are mixed, such as mass, exposure and concentration (please see the attached document for more detail.) The figures are still somehow not well presented and the labels and units are not always clear.

specific comments:

I would like the authors to discuss the sentence "...India manages to sustain the current baseline mortality-GDP relation in future, projected rapid increase in GDP would over-compensate the impact of PM2.5 exposure resulting in a reduction in burden for all the combined scenarios except the ones involving RCP8.5 exposure and SSP3 population." If GDP and Population would grow rapidly would the underlying GHG and pollutant emissions still hold? and therefore the PM2.5 concentration levels? what would have to happen in term of energy and resource efficiency and energy demand for this scenario to be plausible?

Please do not use the word coupling any more and clarify that when mentioning to GDP and population these are based on the SSP scenarios. The current version has too much explanation, a shorter and clearer way to refer to the socio-economic drivers will help the readers.

line 105 refers to a larger correlation whereas the graph does not seem to show the same.

Please rephrase the sentence that starts with line 111.

Please check throughout the text the use of symbols and acronyms and always spell them out when first mentioned.

line 169 refers to figure S3, did you mean S4?

Please correct this "Gross Development Product (GDP)"

In Figure 1 please add a line to represent the baseline mortality it will help the reading.

Please add self explanatory labels to the panels in all the figures, especially in figures 2 and 3.

Table S1 needs units.

In table S3 please describe labels such as D3 ...

in line 204 "whereas lowest premature mortality is expected with SSP5-RCP4.5 scenario; " Why would SSP1 that has lower emissions not be the lowest? is this a regional effect of the the ssp5 baseline? please provide some rational for this result.

In line 212 "The lower limit of the range signifies the premature mortality estimated using (mean- 1σ) PM2.5 exposure for each grid obtained from 13 CMIP5 models and the (central value - error) values of the coefficients that were used in the formulation of the relation between GDP and baseline mortality."

there is a big confusion between concepts, it sound as if the coefficients were represented, please rephrase. And provide simpler and clearer explanations throughout the whole text.

Despite the confusing language and the low quality figures and tables, in my opinion this paper has improved. I think this contribution and ideas are very valuable. But the lack of rigour presented in this paper is significant.

Reviewer #2 (Remarks to the Author):

The authors satisfactorily addressed my original comments with a few exceptions:

- 1) Although the text has improved significantly, I still see a need for a review by an editor. I noticed mistakes in lines 125, 169, 391, 408, 426, 429, 439, 442. There are many places where the article "the" is missing. I also noticed some spelling mistakes in the supplement.
- 2) The captions of the figures and tables are still problematic. First, they should all be reviewed for accuracy. I believe the caption of Table S2 is inaccurate. Caption of Figure 3 states "constant baseline", in contradiction with lines 175-176. The two panels in Figure 3 should be labeled as "a" and "b". The units are missing in Figure 4. It is not clear what is shown in Figure S1. It could be one model according to line 103 or several models according to line 106. If it is some ensemble of 13 models, it should be so stated.
- 3) GBD should be expanded for the benefit of the general reader.
- 4) It would be helpful to quantify "highly" in line 392 with an approximate figure.

Reviewer #4 (Remarks to the Author):

As revised, this manuscript provides a valuable and novel addition to the literature. The modeling methods used are state-of-the-art, and are built on well documented scientific underpinnings. They authors have done a satisfactory job of addressing the prior critiques, with the notable exception that they have not, as requested, provided brief descriptions of each modeled scenario (e.g. RCP4.5 and RCP8.5) in the main paper. The Supplementary Material explanations are fine, but a brief summary of each option is also needed in the main body of the text to help the reader get an appreciation as to the range of options being considered, without searching through the Supplemental Material.

Overall, what is still needed in this manuscript is insight into the greater context of the results in both the "Concluding Remarks" and "Abstract". Comparisons are needed to prior work looking at the clean air mortality benefits of climate mitigation action in India and around the globe (e.g., see West et al, 2013, in Nature Climate Change). Also, please make reference to the 2015 Lancet Commission on Climate and Health conclusions. That would strengthen both the conclusions and Abstract. In that report, the Commission concluded that, as shown in this work, if climate mitigation measures focus on maximizing the human health improvements (e.g., from cleaner air), then considerable local and immediate public health benefits can be also derived from climate mitigation measures, in addition to the climate benefits, if they are directed towards that end (e.g., by focusing on lowering fossil fuel combustion). The Abstract's final sentence (The results are useful for formulating policy that is co-beneficial for mitigating climate change and air pollution.) is a start, but is far too nebulous and weak, given the striking results contained in this manuscript. Something more direct is needed to be said, like: "These results show that, irrespective of their climate benefits, climate mitigation measures can lead to substantial immediate and local clean air health benefits for nations that take those measures, consistent with the 2015 Lancet Commission's conclusion that taking action against climate change can be the "greatest public health opportunity of the 21st Century".

Reviewers' comments:

Reviewer #1 (Remarks to the Author):

I would like to congratulate the authors for the better organization, for the clearer methodology and for the new sensitivity analysis.

However I still think this paper is still not clear enough, I think a mathematical description of the methodology will help the rigour that is lacking. The paper still needs better wording, some sentences seem to be detached from context and above all it needs more rigour, some concepts are mixed, such as mass, exposure and concentration (please see the attached document for more detail.) The figures are still somehow not well presented and the labels and units are not always clear.

We thank the reviewer for providing important suggestions which helped us to improve the manuscript substantially. As suggested, we have mathematically described the methodology in the supplementary material. Lines 92-125

We have attempted to refine the language and use clearer explanations, we now have uniformly used 'exposure' throughout the manuscript. We have modified the figures and legends where ever required.

specific comments:

I would like the authors to discuss the sentence "...India manages to sustain the current baseline mortality-GDP relation in future, projected rapid increase in GDP would over-compensate the impact of PM2.5 exposure resulting in a reduction in burden for all the combined scenarios except the ones involving RCP8.5 exposure and SSP3 population." If GDP and Population would grow rapidly would the underlying GHG and pollutant emissions still hold? and therefore the PM2.5 concentration levels? what would have to happen in term of energy and resource efficiency and energy demand for this scenario to be plausible?

We thank the reviewers for raising this concern. We would like to clarify that we have used the RCP4.5 and 8.5 scenario for estimating the PM_{2.5} exposure and population and GDP that were used in 5 SSPs. We note that SSP emissions were not used in this study. Although the concern raised by the reviewer is very rational, responding to it requires a separate modelling study which is beyond the scope of this work.

Please do not use the word coupling any more and clarify that when mentioning to GDP and population these are based on the SSP scenarios. The current version has too much explanation, a shorter and clearer way to refer to the socio-economic drivers will help the readers.

We thank the reviewer for this. We have refrained from using the word 'coupled' any further in this manuscript. We have attempted to make the descriptions crisp and short.

line 105 refers to a larger correlation whereas the graph does not seem to show the same.

We have addressed the issue in line 99-100

Please rephrase the sentence that starts with line 111.

We have rephrased the line. Which now appears as line 110-111

Please check throughout the text the use of symbols and acronyms and always spell them out when first mentioned.

We thank the reviewer for this. We have thoroughly checked the manuscript for symbols and have spelled out any acronym when first used.

line 169 refers to figure S3, did you mean S4?

We thank the reviewer for this. We have corrected it. We actually meant Supplementary Figure 4 as correctly pointed out by the reviewer

P.S.: (to comply with Nature Communication format we have now changed the format of numbering the supplementary material tables and figures, they are now numbered as Supplementary Table 1/Supplementary Fig 1).

Please correct this "Gross Development Product (GDP)"

We thank the reviewer for pointing this out. We have now spelled GDP correctly as 'Gross Domestic Product'

In Figure 1 please add a line to represent the baseline mortality it will help the reading.

We thank the reviewer for raising this concern. We have added a bold line in Figure 1 that represents baseline exposure.

Please add self explanatory labels to the panels in all the figures, especially in figures 2 and 3.

We have added self explanatory labels the figures 2 and 3

Table S1 needs units.

We have added units to Table 1

In table S3 please describe labels such as D3 ...

The labels have been described now in table caption in the supplementary table 3.

in line 204 "whereas lowest premature mortality is expected with SSP5-RCP4.5 scenario; " Why would SSP1 that has lower emissions not be the lowest? is this a regional effect of the ssp5 baseline? please provide some rational for this result.

We thank the reviewer for raising this concern. We would like to point out that we do not use SSP emissions. We use the population and GDP that were used as drivers for the SSP scenarios. SSP5 may have lower population than SSP1 because of regional effect of SSP5 baseline. We have explained this further in Line 203-206..

In line 212 "The lower limit of the range signifies the premature mortality estimated using (mean-1 σ) PM2.5 exposure for each grid obtained from 13 CMIP5 models and the (central value - error) values of the coefficients that were used in the formulation of the relation between GDP and baseline mortality."

there is a big confusion between concepts, it sound as if the coefficients were represented, please rephrase. And provide simpler and clearer explanations throughout the whole text.

We thank the reviewer for pointing this out. We have rephrased the description accordingly. It appears in lines 210-212.

We have attempted to provide clearer explanations in the manuscript.

Despite the confusing language and the low quality figures and tables, in my opinion this paper has improved. I think this contribution and ideas are very valuable. But the lack of rigour presented in this paper is significant.

We thank the reviewer for identifying our work as valuable. We have attempted to make the paper more rigorous and lucid.

Reviewer #2 (Remarks to the Author):

The authors satisfactorily addressed my original comments with a few exceptions:

1) Although the text has improved significantly, I still see a need for a review by an editor. I noticed mistakes in lines 125, 169, 391, 408, 426, 429, 439, 442. There are many places where the article "the" is missing. I also noticed some spelling mistakes in the supplement.

We thank the reviewer for pointing this out. We have checked for the grammar rigorously. We have made corrections and have rephrased the lines that have been pointed out by the reviewer.

2) The captions of the figures and tables are still problematic. First, they should all be reviewed for accuracy. I believe the caption of Table S2 is inaccurate. Caption of Figure 3 states "constant baseline", in contradiction with lines 175-176. The two panels in Figure 3 should be labeled as "a" and "b". The units are missing in Figure 4. It is not clear what is shown in Figure S1. It could be one model according to line 103 or several models according to line 106. If it is some ensemble of 13 models, it should be so stated.

We thank the reviewer for this. We have checked for the accuracy of the figure and table captions. We have changed and corrected caption for Figure 3. The two panels in Figure 3 are now marked a and b. We have now added unit in Figure 4 (in million). We have now modified the description of Supplementary Fig 1 in the main text Lines 100-101.

3) GBD should be expanded for the benefit of the general reader.

We have now expanded GBD is Global Burden of Diseases when first mentioned.

4) It would be helpful to quantify "highly" in line 392 with an approximate figure.

We have quantified it in Lines 388

Reviewer #4 (Remarks to the Author):

As revised, this manuscript provides a valuable and novel addition to the literature. The modeling methods used are state-of-the-art, and are built on well documented scientific underpinnings. They authors have done a satisfactory job of addressing the prior critiques, with the notable

exception that they have not, as requested, provided brief descriptions of each modeled scenario (e.g. RCP4.5 and RCP8.5) in the main paper. The Supplementary Material explanations are fine, but a brief summary of each option is also needed in the main body of the text to help the reader get an appreciation as to the range of options being considered, without searching through the Supplemental Material.

We thank the reviewer for such detailed and helpful reply. We have now added brief description about RCPs in the main paper lines 149-153.

Detailed description about RCP4.5 and RCP8.5 are provided in supplementary material lines 66-90

Overall, what is still needed in this manuscript is insight into the greater context of the results in both the “Concluding Remarks” and “Abstract”. Comparisons are needed to prior work looking at the clean air mortality benefits of climate mitigation action in India and around the globe (e.g., see West et al, 2013, in Nature Climate Change). Also, please make reference to the 2015 Lancet Commission on Climate and Health conclusions. That would strengthen both the conclusions and Abstract. In that report, the Commission concluded that, as shown in this work, if climate mitigation measures focus on maximizing the human health improvements (e.g., from cleaner air), then considerable local and immediate public health benefits can be also derived from climate mitigation measures, in addition to the climate benefits, if they are directed towards that end (e.g., by focusing on lowering fossil fuel combustion). The Abstract's final sentence (The results are useful for formulating policy that is co-beneficial for mitigating climate change and air pollution.) is a start, but is far too nebulous and weak, given the striking results contained in this manuscript. Something more direct is needed to be said, like: “These results show that, irrespective of their climate benefits, climate mitigation measures can lead to substantial immediate and local clean air health benefits for nations that take those measures, consistent with the 2015 Lancet Commission’s conclusion that taking action against climate change can be the “greatest public health opportunity of the 21st Century”.

We thank the reviewer heartily for this constructive suggestion. We have added discussions in the Concluding remarks section from the Lancet Commission report 2015. Lines 325-330.

We acknowledge that the line previously in Abstract ‘The results are useful for formulating policy that is co-beneficial for mitigating climate change and air pollution’ was weak and nebulous henceforth we removed it and added the first part of the line as suggested by the reviewer to strengthen and conclude the Abstract (Lines 38-40). As we did not examine other countries, however, we did not feel comfortable concluding that other countries could do the same as India.

As suggested by the reviewer, we have compared our study with previous studies such as Silva et al., 2016 and West et al., 2013 in lines 212-218 and 263-265

Reviewers' Comments:

Reviewer #1:

Remarks to the Author:

I acknowledged the effort from the authors, but somehow this paper is still very hard to read and understand in my opinion.

The new mathematical description helps, but the equations and the symbols used are not standard, and make the reading of simple equations too complex. This distracts the readers, equations are meant to help not to keep the reader even more confused.
Please rephrase and rethink the mathematical representation of your problem. ('bl' is not defined)

"mortality-GDP relation" should be shortly explained before, when it is first mentioned.

Regarding my previous comment:

"in line 204 "whereas lowest premature mortality is expected with SSP5-RCP4.5 scenario; " Why would SSP1 that has lower emissions not be the lowest? is this a regional effect of the SSP5 baseline? please provide some rationale for this result.

We thank the reviewer for raising this concern. We would like to point out that we do not use SSP

emissions. We use the population and GDP that were used as drivers for the SSP scenarios. SSP5 may

have lower population than SSP1 because of regional effect of SSP5 baseline. We have explained this

further in Line 203-206.. "

I think this has not been satisfactorily explained, please clarify this better.

I still think the tables (as in table 2) could be presented in a better organization.

Figures 4, S3 and S4 have very poor quality.

There are still no labels in some of the SI tables.

Please harmonize, 'figure' and 'fig' in all the texts.

'IIASA GDP_v9_130219' is not a model

label of the Table S4 is not understandable, and with all the colours the shaded column is not visible.

In figure S5 SSP1 is not visible, please explain in case it is similar to other SSP.

Please proof read this article with a native English speaker, and rephrase sentences such as "

We use population which were used as drivers to develop these scenarios. The storyline which were followed to describe the population for each of the SSP scenarios are described in details in a recent article³"

Reviewer #4:

Remarks to the Author:

The authors have responded to the prior comments well, and the article now provides a novel and valuable addition to the literature. However, the authors note that "household sources are also

major sources of ambient pollution." (which is largely biomass burning) in India, but this is not sufficiently acknowledged in the discussion of the limitations of the analysis, with regard to the applicability of the Relative Risk estimates employed, which are based on western studies for areas dominated by fossil fuel combustion (i.e., not like India). They are also primarily associated with the effect of that type of western PM_{2.5} mass effects on cardiovascular disease. In contrast, recent available cohort type studies of the mortality effects of biomass burning pollution (a dominant source in the India pollution, unlike in the West) have not found similar CVD mortality effects by biomass burning particles (see, e.g., 1.) Alam DS, Chowdhury MA, ... Niessen LW. Adult cardiopulmonary mortality and indoor air pollution: a 10-year retrospective cohort study in a low-income rural setting. *Glob Heart*. 2012 Sep;7(3):215-21. ; and, 2.) Mitter SS, Vedanthan R, ... Malekzadeh R. Household Fuel Use and Cardiovascular Disease Mortality: Golestan Cohort Study. *Circulation*. 2016 Jun 14;133(24):2360-9.). This indicates that the mortality effect of India PM_{2.5} is likely significantly lower on a per ug/m³ basis than that found in past studies in the West (which have predominantly fossil fuel combustion PM_{2.5} sources, instead). As such, the authors need to add another caveat about the uncertainty in the mortality per ug/m³ assumption, citing these studies, and acknowledging the very different composition of PM_{2.5} in India, as opposed to the PM_{2.5} in the air pollution cohort studies relied upon by Burnett et al (Ref. 22). Similarly, for the same reason, I question the statement in the text comparing this situation to London (As a result, the total health burden per capita is likely higher than in London, which started its control of ambient pollution well after it had given up open fires for cooking, although still using coal for space heating. Arguably, therefore the urgency is higher now in India.), since London used coal for the household heating and electricity generation, which past evidence would indicate to have been far more toxic per µg/m³ than biomass burning PM_{2.5}. This comparison to London is not supportable, and should be removed.

Reviewer #1 (Remarks to the Author):

I acknowledged the effort from the authors, but somehow this paper is still very hard to read and understand in my opinion.

The new mathematical description helps, but the equations and the symbols used are not standard, and make the reading of simple equations too complex. this distracts the readers, equation are meant to help not to keep the reader even more confused.

Please rephrase and rethink the mathematical representation of your problem. ('bl' is not defined)

As per the editorial suggestion, we have moved some of the equations in the main text. The subscripts and superscripts in the equation are required to avoid confusion about the source and representativeness of the particular data. We have rephrased the discussion with clarity about each term in the equation preceded by a simple discussion of the method so that the mathematical part can be easily understood. each 'subscript' and 'superscript' is now defined properly.

"mortality-GDP relation" should be shortly explained before, when it is first mentioned.

We thank the reviewer for pointing this out. We have now explained it briefly in lines 115-116.

Regarding my previous comment:

"in line 204 "whereas lowest premature mortality is expected with SSP5-RCP4.5 scenario;

" Why

would SSP1 that has lower emissions not be the lowest? is this a regional effect of the ssp5 baseline? please provide some rational for this result.

We thank the reviewer for raising this concern. We would like to point out that we do not use SSP emissions. We use the population and GDP that were used as drivers for the SSP scenarios. SSP5 may

have lower population than SSP1 because of regional effect of SSP5 baseline. We have explained this

further in Line 203-206.. "

I think this has not been satisfactory explained, please clarify this better.

As earlier mentioned, we use population and GDP that were used as drivers for formulating the SSP scenarios to estimate premature mortality. We do not use the SSP emissions.

Though the exposed (>25yrs) SSP5 population is slightly lower than the SSP1 population over India, the SSP5 GDP is significantly higher than SSP1 GDP which results in a lower baseline mortality (it can be clearly seen in Supplementary material Figure 4).

Table R1: SSP1 and SSP5 population and GDP.

	2010	2020	2030	2040	2050	2060	2070	2080	2090	2100
Population (in millions)										
SSP1	501.9	771.3	928.7	1072.6	1170.8	1222.9	1226.1	1181.9	1103.3	1002.4
SSP5	501.9	770.9	927.5	1070.6	1167.9	1219.3	1221.9	1177.6	1099.2	998.6
GDP (billion US \$ 2005 /yr), IIASA										
SSP1	3885.4	10511.4	21326.2	31373.3	39869.2	36770.9	51847.8	56316.6	59856.6	61624.9
SSP5	3885.4	10603.6	22408.7	34960.7	46993.5	58592.4	71507.7	85215.6	97649.2	106999

As both GDP and exposed population of SSP5 are lower than SSP1, the premature mortality burden is lower. Though the SSP1 emission is less than SSP5 emission, as we do not consider SSP emissions, this does not alter our results. The lower population and GDP for India for SSP5 might be a regional effect.

I still think the tables (as in table 2) could be presented in a better organization.

We have now attempted to represent it in a better way. We have removed the colors from the tables to make them clear and readable.

Figures 4, S3 and S4 have very poor quality.

Figure 4 is now Table 1. Since now Table 1 and Supplementary table 2 have similar information, we removed the Supplementary table 2 and only kept Table 1 where premature mortality estimate for each decade is given for all 10 combined scenarios. We have improved the quality of Supplementary material Figure 3 and 4.

There are still no labels in some of the SI tables.

All the supplementary tables were checked for labels.

Please harmonize, 'figure' and 'fig' in all the texts.

It is now harmonized as figure throughout the text

'IIASA GDP_v9_130219' is not a model

We thank the reviewer for this. It has now been rectified

label of the Table s4 is not understandable, and with all the colours the shaded column is not visible.

The colours are now removed for better understanding

In figure s5 ssp1 is not visible, please explain in case it is similar to other ssp.

SSP1 overlaps with SSP5 as the SSP5 and SSP1 exposed population are almost similar. (Table R1, in this document)

Please proof read this article with a native English speaker, and rephrase sentences such as "

We use population which were used as drivers to develop these scenarios. The storyline which were followed to describe the population for each of the SSP scenarios are described in details in a recent articles3"

The article is proof read by KRS who is a native English speaker. Several sentences are modified to enhance clarity.

Reviewer #4 (Remarks to the Author):

The authors have responded to the prior comments well, and the article now provide a novel and valuable addition to the literature. However, the authors note that "household sources are also major sources of ambient pollution." (which is largely biomass burning) in India, but this is not sufficiently acknowledged in the discussion of the limitations of the analysis, with regard to the applicability of the Relative Risk estimates employed, which are based on western studies for areas dominated by fossil fuel combustion (i.e., not like India). They are also primarily associated with the effect of that type of western PM2.5 mass effects on cardiovascular disease. In contrast, recent available cohort type studies of the mortality effects of biomass burning pollution (a dominant source in the India pollution, unlike in the West) have not found similar CVD mortality effects by biomass burning particles (see, e.g., 1.) Alam DS, Chowdhury MA, ... Niessen LW. Adult cardiopulmonary mortality and indoor air pollution: a 10-year retrospective cohort study in a low-income rural setting. *Glob Heart*. 2012 Sep;7(3):215-21. ; and, 2.) Mitter SS, Vedanthan R, ... Malekzadeh R. Household Fuel Use and Cardiovascular Disease Mortality: Golestan Cohort Study. *Circulation*. 2016 Jun 14;133(24):2360-9.). This indicates that the mortality effect of India PM2.5 is likely significantly lower on a per ug/m3 basis than that found in past studies in the West (which have predominantly fossil fuel combustion PM2.5 sources, instead). As such, the authors need to add another caveat about the uncertainty in the mortality per ug/m3 assumption, citing these studies, and acknowledging the very different

composition of PM_{2.5} in India, as opposed to the PM_{2.5} in the air pollution cohort studies relied upon by Burnett et al (Ref. 22).

We thank the reviewer for such exhaustive comments that help improving the previous versions of our manuscript. The IERs consider household exposure in their formulation. However, we agree that it only depends on mass, not the composition. The issue of PM_{2.5} composition on relative risk is an active field of research. We cited these two references in the "methods" section under the subheading of 'underlying assumptions'. We already pointed out the issue of contribution of household sources on ambient PM_{2.5} concentration in India (lines 511-514).

Similarly, for the same reason, I question the statement in the text comparing this situation to London (As a result, the total health burden per capita is likely higher than in London, which started its control of ambient pollution well after it had given up open fires for cooking, although still using coal for space heating. Arguably, therefore the urgency is higher now in India.”), since London used coal for the household heating and electricity generation, which past evidence would indicate to have been far more toxic per µg/m³ than biomass burning PM_{2.5}. This comparison to London is not supportable, and should be removed.

This comparison is valid at least in the cities (like Delhi) where the major source is fossil-fuel combustion. We agree that in rural areas dominated by biomass burning PM_{2.5}, this comparison may not hold true. We have modified the discussion in this section.